# Joint control of meiotic crossover patterning by the synaptonemal complex and HEI10 dosage

Stéphanie Durand[1,4], Qichao Lian [1,4], Juli Jing [1,4], Marcel Ernst [2], Mathilde Grelon[3], David Zwicker [2] & Raphael Mercier [1] ✉

Meiotic crossovers are limited in number and are prevented from occurring close to each other by crossover interference. In many species, crossover number is subject to sexual dimorphism, and a lower crossover number is associated with shorter chromosome axes lengths. How this patterning is imposed remains poorly understood. Here, we show that overexpression of the Arabidopsis pro-crossover protein HEI10 increases crossovers but maintains some interference and sexual dimorphism. Disrupting the synaptonemal complex by mutating *ZYP1* also leads to an increase in crossovers but, in contrast, abolishes interference and disrupts the link between chromosome axis length and crossovers. Crucially, combining HEI10 overexpression and *zyp1* mutation leads to a massive and unprecedented increase in crossovers. These observations support and can be predicted by, a recently proposed model in which HEI10 diffusion along the synaptonemal complex drives a coarsening process leading to well-spaced crossover-promoting foci, providing a mechanism for crossover patterning.

A hallmark of sexual reproduction is the shuffling of homologous chromosomes by meiotic crossovers (COs). COs are produced by the repair of DNA double-strand breaks through two biochemical pathways: Class I COs are produced by a meiotic-specific pathway catalyzed by the ZMM proteins (*Saccharomyces cerevisiae* Zip1-4, Msh4-5, and Mer3; HEI10 is the Arabidopsis homolog of Zip2) and represent most COs; Class II COs originate from a minor pathway that uses structure-specific DNA nucleases also implicated in DNA repair in somatic cells. Despite an excess of initial double-strand breaks at meiosis, the number of resulting COs is limited, typically to one to three per chromosome pair. Class I COs are subject to additional tight constraints: At least one class I CO occurs per chromosome pair at each meiosis, the so-called obligate CO that ensures balanced chromosome distribution. Class I COs are also prevented from occurring next to each other on the same chromosome, a phenomenon called CO interference. How this interference is achieved mechanistically has been debated for over a century[1–6].

One specific unresolved question is the role of the synaptonemal complex (SC) in CO interference. The SC is a zipper-like tripartite structure composed of two lateral chromosome axes, along which arrays of chromatin loops of each of the two homologous chromosomes are anchored, and a central part consisting of transverse filaments that connect the axes all along their length at meiotic prophase. Assessing the role of the SC in interference is difficult because in many organisms the transverse filament protein is essential for the formation of class I COs[6]. One notable exception is *Arabidopsis thaliana*, where the transverse filament protein is not required for class I CO formation, providing a unique opportunity to analyze the role of the SC in CO patterning. In the *zyp1* mutant, class I COs form at a higher frequency than wild type and completely lack interference, demonstrating that the central part of the SC is, directly or indirectly, essential for imposing CO interference in Arabidopsis[7,8]. Reduced expression of the transverse element in *C. elegans* and specific mutations of the SC

[1]Department of Chromosome Biology, Max Planck Institute for Plant Breeding Research, Carl-von-Linné-Weg 10, 50829 Cologne, Germany. [2]Max Planck Institute for Dynamics and Self-Organization, Am Faßberg 17, 37077 Göttingen, Germany. [3]Université Paris-Saclay, INRAE, AgroParisTech, Institut Jean-Pierre Bourgin (IJPB), 78000 Versailles, France. [4]These authors contributed equally Stéphanie Durand, Qichao Lian, Juli Jing. ✉e-mail: mercier@mpipz.mpg.de

component that uncouple SC and CO formation in budding yeast lead to a reduction of interference, supporting a conserved role of the SC in imposing interference[9–12]. Interestingly in some species, such as humans and Arabidopsis, CO number differs in males and females. This heterochiasmy correlates with axis/SC length, with the number of COs proportional to axis length[13–15]. CO interference appears to propagate at a similar axis/SC distance (μm) in both sexes, which means that interference acts over greater genomic ranges (DNA) in the sex with a shorter axis/SC[15,16], an observation that shows that the relevant space for the mechanism of interference is the axis/SC length.

A model was recently elaborated to account for class I CO patterning and interference, based on diffusion of the ZMM protein HEI10 (ZHP-3/4 in *C.elegans*) within the SC and a coarsening process leading to well-spaced CO-promoting HEI10 foci[17,18]. HEI10, which encodes a E3 ubiquitin ligase, initially forms multiple small foci along the SC and is progressively consolidated into a small number of large foci that co-localize with CO sites in diverse species[19–22]. Further, as predicted by the model, CO numbers depend on HEI10 dosage in Arabidopsis[18,23]. Interference is abolished in the absence of the transverse filament of the synaptonemal complex ZYP1[7,8], which is compatible with the idea that diffusion of HEI10 along the central part of the SC underlies CO patterning and interference.

Here, we explore the mechanisms of CO patterning in Arabidopsis by analyzing the combinatory effects of axis/SC length (male *vs.* female), modification of HEI10 dosage, and disruption of the SC on COs. We show that overexpressing HEI10 in *zyp1* completely deregulates class I COs, with a massive increase of their number in both females and males. Our results support the model in which HEI10 coarsening by diffusion along the SC mediates CO patterning and imposes CO interference.

## Results and discussion
### Experimental approach

To decipher CO control, we studied the number and distribution of COs in both female and male meiosis when overexpressing HEI10 (well-characterized C2 line[23]), in the absence of the synaptonemal complex (*zyp1*), and in combination. We measured the number of class I COs in meiocytes by counting the number of MLH1-HEI10 co-foci at diplotene (Fig. 1A, B, Supplementary Fig. 1). In the pure line Col, we analyzed six genotypes: wild type and *zyp1-1* combined with three dosages or HEI10 (wild type, heterozygous or homozygous for the HEI10[oe] C2 transgene). In the Col/L*er* F1 hybrid, we analyzed four genotypes: wild type and *zyp1-1/zyp1-6* combined with two dosages of HEI10 (wild type and heterozygous for the C2 HEI10 transgene) (Fig. 1A, B). The four hybrid genotypes were also used to characterize the CO number and distribution by sequencing populations derived from female and male crosses to Col (Figs. 1C–F, 2 and Supplementary Fig. 2–10, Supplementary Data 1).

### Overexpression of HEI10 increases COs but maintains heterochiasmy

In wild type, the number of MLH1 foci is higher in males than females in both the inbreds and the hybrids (ratio male/female = 1.8 and 1.6, respectively). Figure 1A, B. Whole-genome sequencing of male- and female-derived hybrid progenies showed that CO numbers detected genetically are higher in male meiosis than in female meiosis (Fig. 1D, ratio = 1.6, *p* < 0.001, Supplementary Figs. 2 and 3), confirming heterochiasmy. The number of MLH1 foci at male meiosis is higher in wild type Col than in Col/L*er*. Analysis of quantitative trait loci (QTL) in a Col/L*er* population previously revealed that the Col *HEI10* allele is associated with higher recombination levels, suggesting that at least a part of this difference in MLH1 counts can be attributed to a difference in HEI10 activity[23]. In wild type female, the MLH1 foci numbers are not significantly different between Col and Col/L*er* and close to the minimum of one per chromosome (7.2 and 6.8 foci for five chromosomes).

In the presence of a transgene ectopically overexpressing HEI10 (HEI10[oe] C2 line[23], homozygous), the number of MLH1 foci is increased ~two-fold in both sexes, in both Col and Col/L*er*. Heterozygosity for the HEI10[oe] transgene also increases MLH1 foci number, but slightly less than homozygosity, confirming the effect of HEI10 dosage on recombination[18,23] and suggesting that the level of HEI10 in the C2 line is close to saturation. Importantly, increases provoked by HEI10 dosage modulation are similar in males and females, leading to more MLH1 foci in males than females (*p* = 0.0001) (Fig. 1B). This was confirmed with progeny sequencing in hybrids, which revealed a 2.1-fold increase of COs in HEI10[oe] female and male, compared with wild type (*p* < 10^-15, Fig. 1D–F, Supplementary Fig. 3). The ratio of male *vs.* female COs is maintained at 1.6 in HEI10[oe] (*p* < 10^-15). In summary, over-expressing HEI10 provokes a doubling of class I COs in both female and male, maintaining heterochiasmy.

### *ZYP1* mutation increases COs and abolishes heterochiasmy

Mutating the transverse filament of the SC ZYP1 also increases MLH1 foci number (Fig. 1B). In Col *zyp1*, compared to wild type, the numbers increased 1.4-fold in males, consistent with previous findings[7,8], and 2.3-fold in females. In the Col/L*er* hybrid, the numbers increased by 1.2-fold in male and 1.8-fold in females. In contrast to HEI10[oe], MLH1 foci number, which marks specifically class I COs, is no longer significantly different in males versus females both in Col and Col/L*er* (*p* > 0.6). This is consistent with genetic COs detected by sequencing of hybrid progenies with equal numbers observed in the chromosome sets transmitted by female and male gametes, and fold increases of 2.3 in females and 1.5 in males compared to wild type (Fig. 1D, Supplementary Fig. 3)[8]. The *zyp1* mutation thus leads to an increase in class I COs, which disproportionately affects female meiosis and abolishes heterochiasmy.

### Combining HEI10 overexpression and *zyp1* massively increases class I COs

HEI10[oe] and *zyp1* increase CO number, but in different ways; while the former maintains heterochiasmy and some interference, the latter does not. We thus combined *zyp1* mutation and HEI10[oe] and analyzed the effects on MLH1 foci numbers (Fig. 1A, B). In Col, the number of foci observed in *zyp1* mutants homozygous for the HEI10[oe] transgene was significantly higher than ever previously reported, reaching 47.8 and 45.0 in females and males, respectively. The female and male MLH1 counts are not significantly different from each other and represent marked 6.7-fold and 3.5-fold increases compared to their respective wild types. In Col *zyp1* males heterozygous for HEI10[oe], the MLH1 foci count was slightly but significantly lower (41.1, *p* = 0.015) than the homozygous, showing that there is a dynamic range of HEI10 dosage effects on COs. In the hybrid *zyp1* HEI10[oe], the observed number of MLH1 in females and males was 29.8 and 30.0, not significantly different from each other (*p* = 0.8) but representing a 4.4- and 2.9-fold increase compared to their wild type controls. This suggests that class I COs are massively increased in *zyp1* mutants overexpressing HEI10. When comparing Col and Col/L*er zyp1* HEI10[oe het], the number of MLH1 foci is higher in the pure line (41.1) than in the hybrid (30). The endogenous HEI10 allele may contribute to this difference, but probably has a limited effect in this context of HEI10 over-expression. Alternatively it is possible that the DNA polymorphisms between homologous chromosomes in the hybrid decrease the number of CO-eligible recombination intermediates.

Progeny sequencing showed that the number of genetic COs in male *zyp1* HEI10[oe] was increased compared to wild type, reaching 14.7 CO per chromatid set (3.1-fold, Mann–Whitney test, *p* < 2.2e−16, Fig. 1D–F), fitting well with the 30 MLH1 foci counted in male meiocytes (Fig. 1B, Supplementary Fig. 11). In females, COs were also increased to even higher levels than predicted by the number of MLH1 foci (30/2 = 15), reaching 19.6 COs per female chromatid set (6.4-fold/wild type,

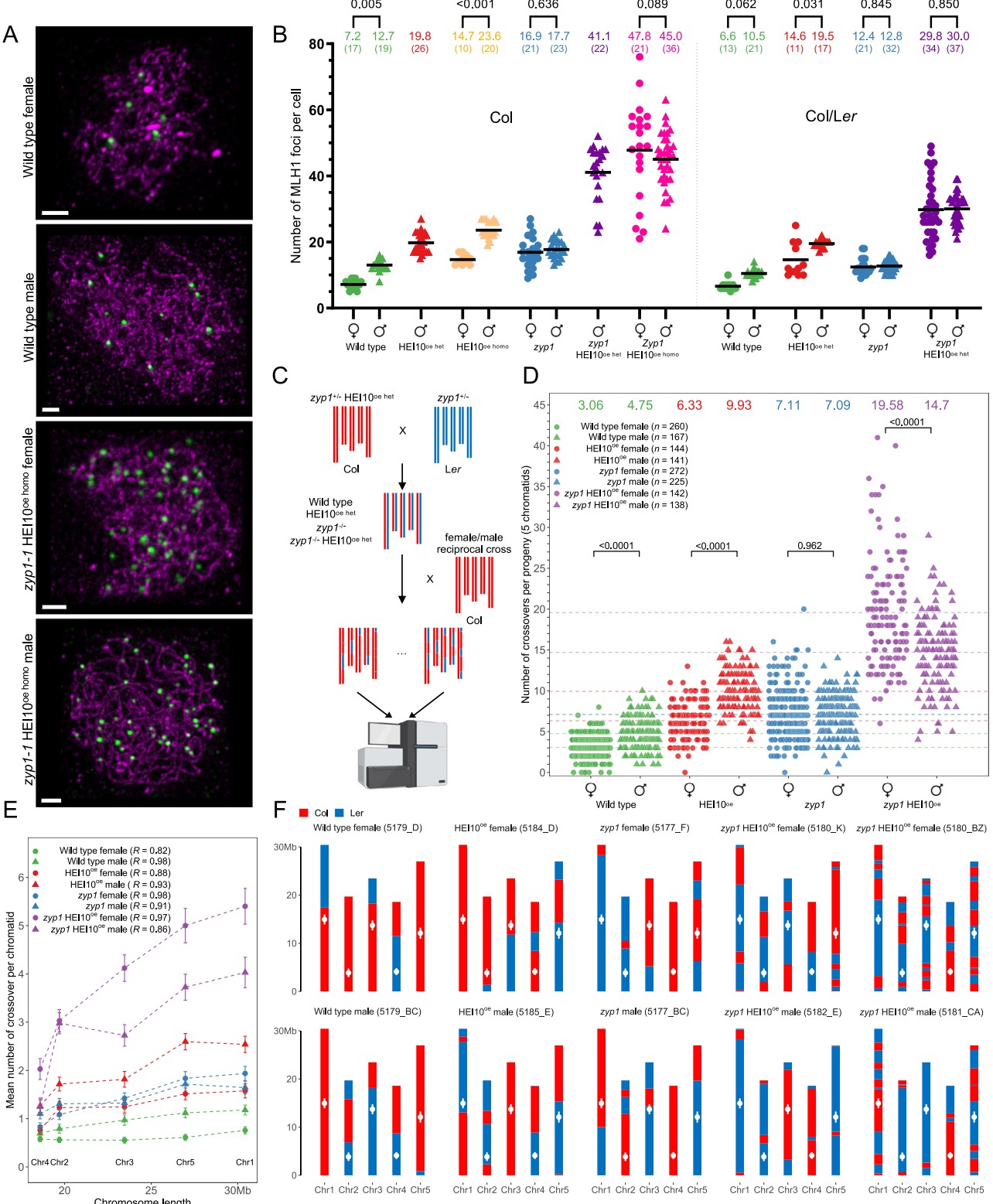

*p* < 10⁻¹⁵, Fig. 1D–F). Together, this shows that combining *zyp1* mutation and HEI10 overexpression increases the numbers of class I COs. The greater number of detected genetic COs compared to what was predicted based on the number of MLH1 foci suggests that class II COs may also be increased in female *zyp1* HEI10ᵒᵉ. One plausible scenario is that in absence of ZYP1, over-expression of HEI10 could protect recombination intermediates from anti-CO helicase activity[24], which would be then repaired as class II COs by nucleases such as MUS81.

While the number of class I CO (MLH1 foci) appear to be identical in both sexes in *zyp1* HEI10ᵒᵉ, a component of heterochiasmy is revealed in this context, with now more CO in females than males, presumably due to a large increase of class II CO in female meiosis. The observed increase in class I COs is unprecedented, and suggests that the tripartite SC and HEI10 levels are two main regulators limiting class I COs.

Looking along chromosomes, *zyp1* and HEI10ᵒᵉ individually or in combination elicit a massive increase in COs along the arms while the

**Fig. 1 | Massive increase in crossovers through combination of _zyp1_ mutation and HEI10 overexpression. A** MLH1 foci in Col wild type and _zyp1_ HEI10[oe homo] meiocytes. Following immunolocalization, REC8 (Purple) and HEI10 (Supplementary Fig. 1) were imaged with STED while MLH1 (green) was imaged with confocal microscopy. The maximum intensity projection is shown. Scale bar = 1 μm. **B** Corresponding MLH1-HEI10 foci quantification, in female and male, inbred Col and hybrid Col/L_er_. The HEI10 transgene originates from the C2 line and is either homozygous (HEI10[oe het]) or heterozygous (HEI10[oe homo]). Each dot is an individual cell,; circles and triangles are females and males, respectively. the mean is indicated by a bar and a number on the top; The number of analyzed cells are indicted into brackets. _P_ values are from one-way ANOVA followed by Fisher's LSD test. **C** Experimental design for construction of female and male hybrid populations for

sequencing. Created with BioRender.com. **D** The number of COs per chromatid set transmitted by female and male gametes of wild type, HEI10[oe], _zyp1_, and _zyp1_ HEI10[oe]. Each point is a BC1/gamete, circles and triangles are females and males, respectively. The means are indicated by horizontal dashed lines and numbers on the top. _P_ values are from one-way ANOVA followed by Fisher's LSD test. The population size is shown in parentheses. **E** Correlation analysis between mean number of COs per transmitted chromatid and chromosome size (Mb). Error bars are the 90% confidence intervals of the mean. Pearson's correlation coefficients are shown in parentheses. The sample sizes, _n_, are identical to panel **D**. **F** Genotypes are shown for representative transmitted chromatid sets in wild type and mutants, and for extreme cases in _zyp1_ HEI10[oe]. Centromere positions are indicated by white points. Source data are provided as a Source Data file.

peri-centromeres and the Col/L_er_ large inversion[25,26] remained recalcitrant to recombination (Fig. 2, Supplementary Fig. 4). A higher density of structural polymorphism appears also to locally limit recombination (Supplementary Fig. 4). At the fine scale, the majority of COs were located in genic regions in both wild type and mutants (Supplementary Fig. 5). This suggests that despite a large increase in CO number, the local preference for CO placement is conserved, presumably because the distribution of double-strand breaks is maintained. For all eight hybrid populations, the average observed number of COs is positively correlated with the physical size (Mb) of chromosomes (Pearson's correlation coefficients >0.8, Fig. 1E). We looked for co-variation of CO frequency between chromosomes within the same meiocyte/gamete, as observed in various species[27]. No significant correlation was seen in any of the populations, with maximum correlation coefficients of -0.2 observed in female _zyp1_ HEI10[oe] (Supplementary Fig. 6), suggesting that this co-variation does not exist in Arabidopsis or is too small to be detected in our assay.

### CO interference is reduced by HEI10 overexpression and abolished in _zyp1_

To measure the impact of _zyp1_ and HEI10[oe] on CO interference, we first analyzed the distribution of distances between two genetically detected COs for chromosomes with exactly two COs in chromatids derived from female and male meiosis (Fig. 2, Supplementary Fig. 7). Note that this approach is to some extend limited as a single chromatid captures only half of the COs occurring on a given bivalent[28] (Supplementary Fig. 11). In wild type females and males, the distribution was significantly shifted to large inter-CO distances ($p < 10^{-6}$) compared with the expected distribution if the COs were randomly spaced, showing the presence of CO interference (Fig. 2C). In HEI10[oe] females and males, the distribution was also shifted to longer distances, showing the presence of CO interference in both sexes ($p < 10^{-4}$, Fig. 2D), consistent with previous results[18,29]. However, the shift was less marked than in the wild type, suggesting a reduction of interference in HEI10[oe]. In _zyp1_ and _zyp1_ HEI10[oe], the observed distributions of inter-CO distances were not different from what would be expected in the case of random spacing ($p > 0.2$, Fig. 2E, F), suggesting an abolition of CO interference in both females and males. Furthermore, we performed a coefficient of coincidence (CoC curve) analysis that accurately describes CO interference[3,30] (Fig. 2G–J, Supplementary Fig. 8). In wild type, the two CoC curves are below 1 at distances < ~15 Mb in both females and males, confirming the presence of substantial CO interference (Fig. 2G). The female curve stays close to 0 for longer distances, showing that CO interference propagates to longer Mb distances in females, consistent with previous analyses[8,14,16]. In HEI10[oe], the curves also deviate from 1 at short distances (<~7 Mb), showing the presence of interference, although at a reduced level compared to wild type (Fig. 2H). As in wild type, interference in HEI10[oe] is stronger in female than in male meiosis. In contrast, the CoC curves are flat at values close to 1 for both females and males in _zyp1_ (Fig. 2I), confirming that CO interference is abolished in the absence of ZYP1[7,8]. In _zyp1_ HEI10[oe], the curves are also flat at ~1, showing that the numerous class I COs

produced in this context do not interfere with each other (Fig. 2J). Thus, HEI10[oe] reduces, while _zyp1_ abolishes CO interference.

### Only mild meiotic defects are observed in _zyp1_ HEI10[oe]

The limited level of COs per chromosome observed in most eukaryotes could suggest that a high level of COs has a detrimental effect. We explored if a massive elevation of class I COs is associated with meiotic chromosome segregation and fertility defects. The number of seeds per fruit is reduced in _zyp1-1_ compared to wild type (−8%, _t_-test, $p < 0.001$), consistent with previous results and the reported loss of the obligate CO in _zyp1_ mutants[7,8] (Fig. 3I). Analyses based on sequence coverage detected a few aneuploids among _zyp1_ transmitted chromatids (2/497, Fig. 3J and Supplementary Figs. 9 and 10) that were not detected in hybrid wild types (0/427 in this study, and 0/760 in an independent wild type dataset[31]). The HEI10[oe] C2 line also showed a slight reduction of fertility (−12%, $p = 0.005$, Fig. 3I) and low frequency of aneuploid chromatid sets (2/285). In _zyp1_ HEI10[oe], seed number was reduced (−7%, $p = 0.025$), and a small number of aneuploids were detected in hybrids (7/272), suggesting a slight meiotic defect also in this background. All the 11 identified trisomy cases concerned chromosome 4, the shortest Arabidopsis chromosome. The frequency of aneuploids we observed among progeny is likely an underestimation of the meiotic missegregation, as it was reported that transmission of gametes with an extra chromosome is low in Arabidopsis[32]. The centromeric region of chromosome 4 of the aneuploids is systematically heterozygous Col/L_er_, which is diagnostic for missegregation at meiosis I (failure to separate homologous chromosomes). For the vast majority (9/11), no COs were detected on the aneuploid chromosome, which is compatible with an absence of COs in the bivalent (however, a transmitted chromatid without a CO can result from a bivalent with CO, Supplementary Fig. 11). None of the aneuploids were among the samples with very high CO numbers (up to seven on a single chromosome 4, Supplementary Fig. 3) This favors the hypothesis that these nine events resulted from the loss of the obligate CO and consequent random missegregation of univalents, rather than an alternative hypothesis in which elevated CO number could disturb chromosome segregation. It should be also noted that a translocation is associated with the HEI10 C2 transgene, which may also contribute to chromosome missegregation[29]. Two aneuploids, both from _zyp1_ HEI10[oe], had two COs on the trisomic chromosome. In both cases, the two COs are relatively close to each other (~2 and 4 Mb), which may lead to an unstable connection between the homologs as spindle tension would be counteracted by only a short stretch of cohesion. Meiotic chromosome spreads in _zyp1_ HEI10[oe] showed that most metaphase I cells had a wild type configuration with five bivalents aligned on the metaphase plate (44/45 in Col; 23/30 in Col/L_er_; Fig. 3C). However, one univalent was observed in a minority of cells (1/45 and 7/30, Fig. 3E). Consistently, at metaphase II almost all cells had five chromosomes aligned on the two plates (25/25 and 6/7; Fig. 3D), and one had a 6:4 configuration indicating unbalanced segregation at meiosis I (Fig. 3F), likely due to the

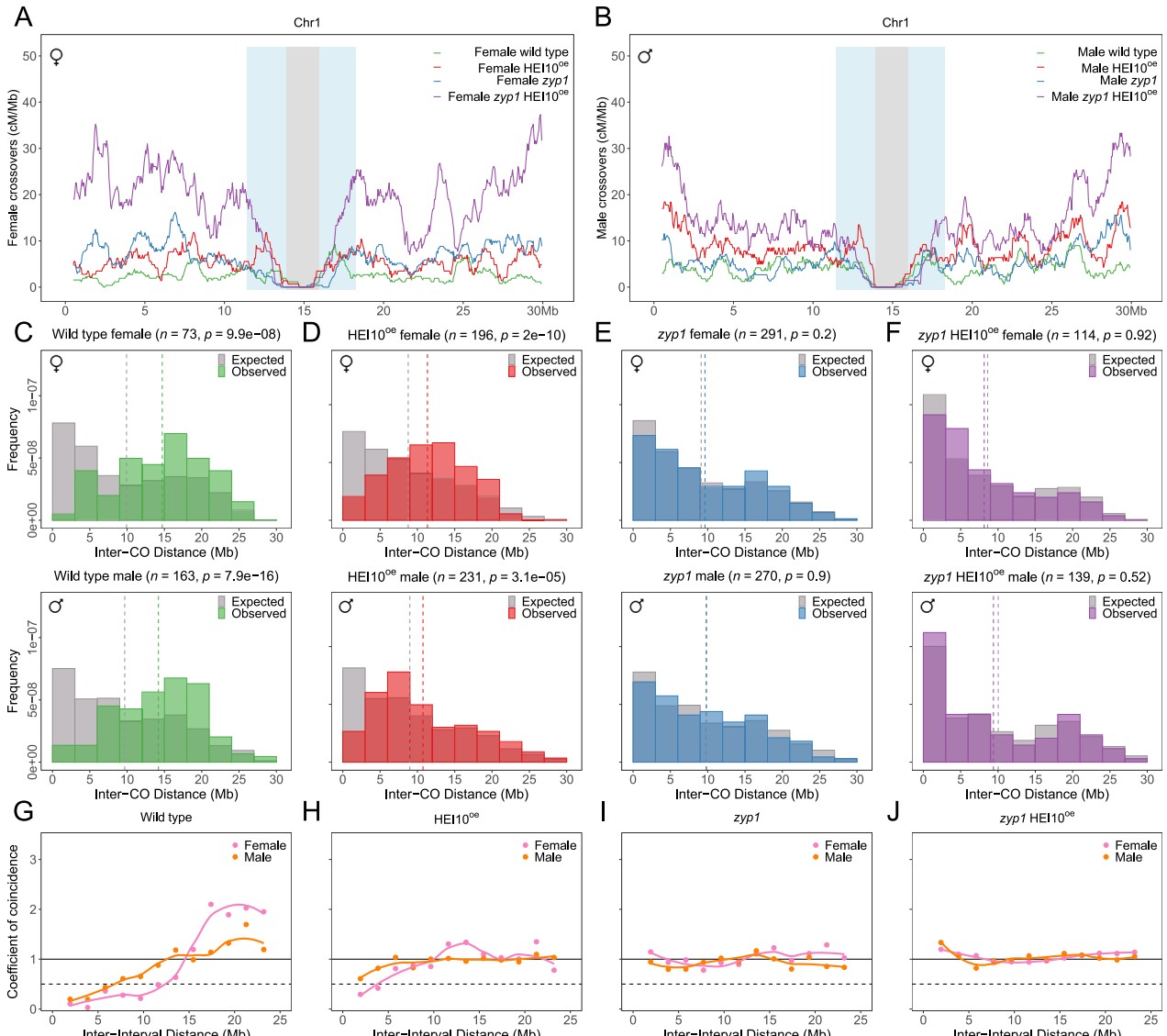

**Fig. 2 | CO distribution and interference analysis in female and male wild type, HEI10^oe, zyp1, and zyp1 HEI10^oe.** The distribution of COs on chromosome 1 in **A** female and **B** male of wild type, HEI10^oe, zyp1, and zyp1 HEI10^oe. The other chromosomes and genomic features are shown in Supplementary Fig. 4. **C**–**F** Distribution of distances between two COs for chromosomes with exactly two COs (Supplementary Fig. 7). The gray bar represents the expected distribution of COs without interference, calculated by permutation analysis of COs (see methods). The number of analyzed CO pairs and the p value from the two-sided Mann–Whitney test between the expected and observed are indicated. **G**–**J** CoC curves in female and male meiosis of wild type, HEI10^oe, zyp1, and zyp1 HEI10^oe, respectively. Chromosomes were divided into 13 intervals, for calculating the mean coefficient of coincidence of each pair of intervals. Source data are provided as a Source Data file.

absence of the obligate CO. Altogether, this shows that a slight meiotic chromosome segregation defect is present in HEI10^oe zyp1. However, the rare missegregations appear to be due to an incomplete CO assurance and do not appear to be associated with the extreme CO numbers observed in the mutants (up to 15 COs in a single chromatid, Supplementary Fig. 3). This suggests that high CO number does not impair chromosome segregation and raises the question of the evolutionary forces that limit CO to typically less than three per chromosome per meiosis in most eukaryotes[4,33]. While failure to ensure at least one CO per chromosome pair is associated with meiotic failure in most eukaryotes, the reasons that prevent high CO numbers are unclear. The absence of an immediate cost of elevated CO numbers in HEI10^oe zyp1 suggests that low CO numbers are not selected for by evolution because of mechanical constraints during meiosis. Rather, this observation suggests that the medium-to-long term genetic effects of COs are subject to indirect selection[4]. This supports the suggestion that a relatively

low recombination rate, not much higher than one per chromosome, is optimal for adaptation.

## Female and male chromosome axis lengths differ and are affected by neither HEI10^oe nor zyp1

SC length has been shown to correlate with the frequency of class I COs[13–15]. We wondered if the class I CO increase provoked by zyp1 and HEI10^oe is associated with variation in SC length. We traced chromosome axes (REC8) in female and male meiocytes with preserved 3D organization and measured the length of each chromosome (Fig. 4). In wild type, we found that the SC is 1.6-fold longer in males than females, consistent with previous reports[14] (Fig. 4M). The longer total SC length in wild type males is proportional to the higher MLH1 foci and CO numbers compared to females (Figs. 1B, 1D and 4Q), suggesting that SC length determines CO number and thus drives heterochiasmy. SC/axis absolute and relative length is conserved in both sexes in HEI10^oe, zyp1, and zyp1 HEI10^oe mutants, thus maintaining the male-female

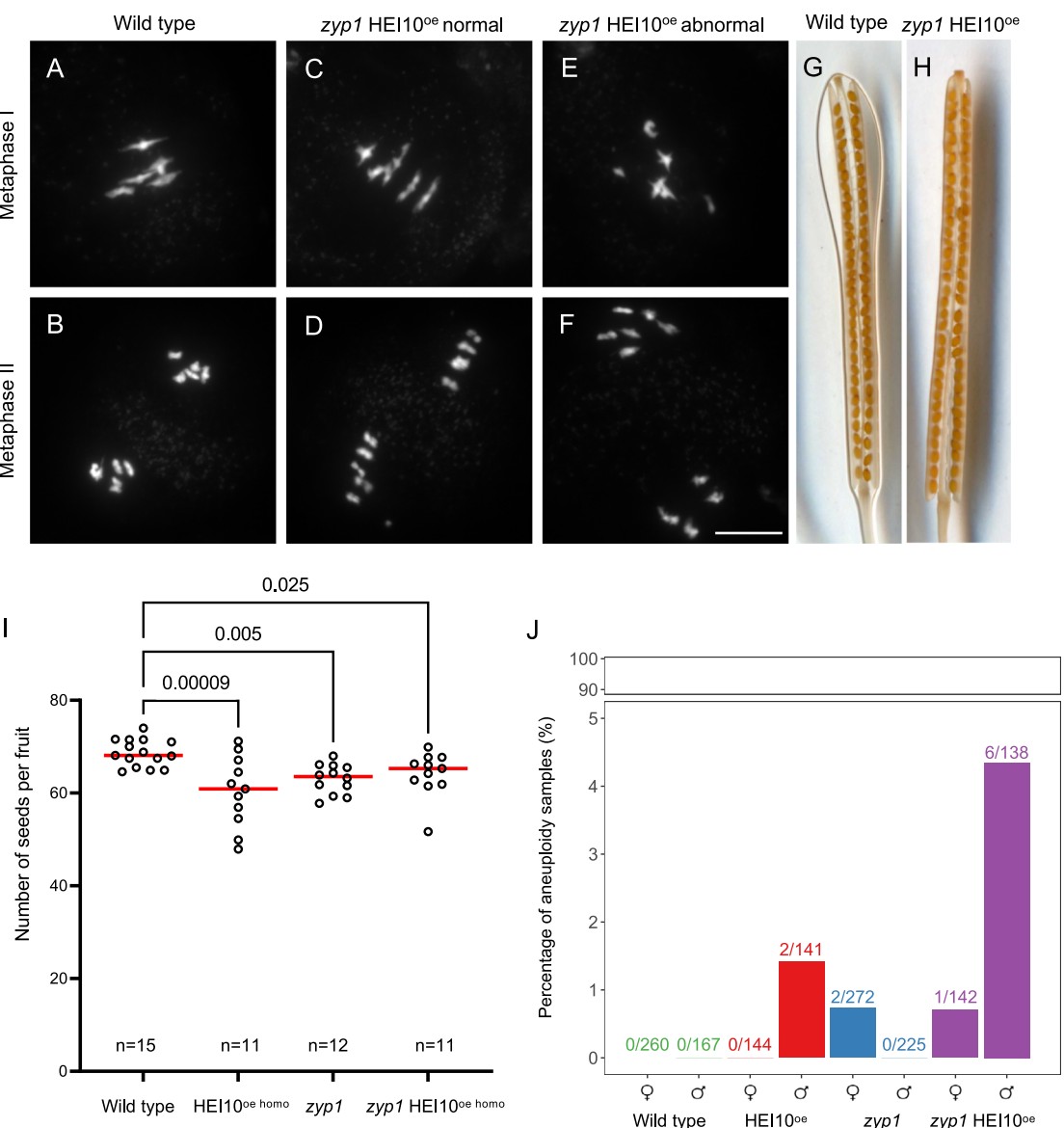

**Fig. 3 | Analysis of meiotic and fertility defects. A–F** DAPI-stained meiotic chromosome spreads from Col/L*er* male meiocytes in wild type (**A, B**) and *zyp1* HEI10oe (**C–F**). **A, C, E** Metaphase I. **B, D, F** Metaphase II. **C, D** Normal chromosome configurations in *zyp1* HEI10oe. **E, F** Rare abnormal chromosome configurations in *zyp1* HEI10oe. Scale bar = 10 μm. **G, H** Representative cleared fruits of wild type Col and *zyp1* HEI10oe mutants. **I** Corresponding quantification of fertility. Each dot represents the fertility of an individual plant, measured as the number of seeds per fruits averaged on ten fruits. The red bar shows the mean. All plants were siblings grown together in a growth chamber. The number n of analyzed plants is indicated and *P* values are one-way ANOVA followed by Fisher's LSD test. **J** The percentage of aneuploid samples detected in each population (Supplementary Figs. 9 and 10). The proportion of aneuploid samples in each population is shown on top of the bars. Source data are provided as a Source Data file.

dimorphism (Fig. 4M). In HEI10oe, the MLH1 foci and CO numbers are increased proportionally in males and females, maintaining heterochiasmy (Figs. 1B and 4O–Q). This suggests that the effect of HEI10 dosage on COs is constrained by the length of the SC. In clear contrast to HEI10oe, the link between axis length and CO number is disrupted in *zyp1*, with MLH1 foci and COs equal in males and females despite a large difference in axis length (4O–Q). The observation that the length of pairs of axes in *zyp1* matches the length of the assembled SC in the wild type suggests that the length of the two axes directly determines SC length. In the double mutant *zyp1* HEI10oe, MLH1 foci are increased and reach equal numbers in males and females despite different axis lengths that are unmodified compared to wild type (Fig. 4P). This suggests that HEI10 dosage has a comparable effect in males and females in the absence of the SC.

Altogether, this suggests that two major factors conjointly regulate CO number: (i) Our results show that the transverse filament of the SC ZYP1 imposes interference and limits COs. The length of the axis/SC is correlated with the number of COs in various contexts[13], and when comparing sexes. Crucially, this correlation is lost in the absence of ZYP1, where the difference in axis length is no longer associated with a difference in CO number, suggesting that COs are regulated by the length of the tripartite SC and thus indirectly by the axis. The upstream mechanisms that determine the differences in SC lengths in males and females in many organisms remain to be determined. (ii) HEI10 dosage positively regulated CO formation. The effect of HEI10 dosage appears to be constrained by the length of the SC. HEI10 initially loads as multiple foci along the SC before consolidating into a small number of large foci at CO sites[19]. This supports a model in which HEI10 loading

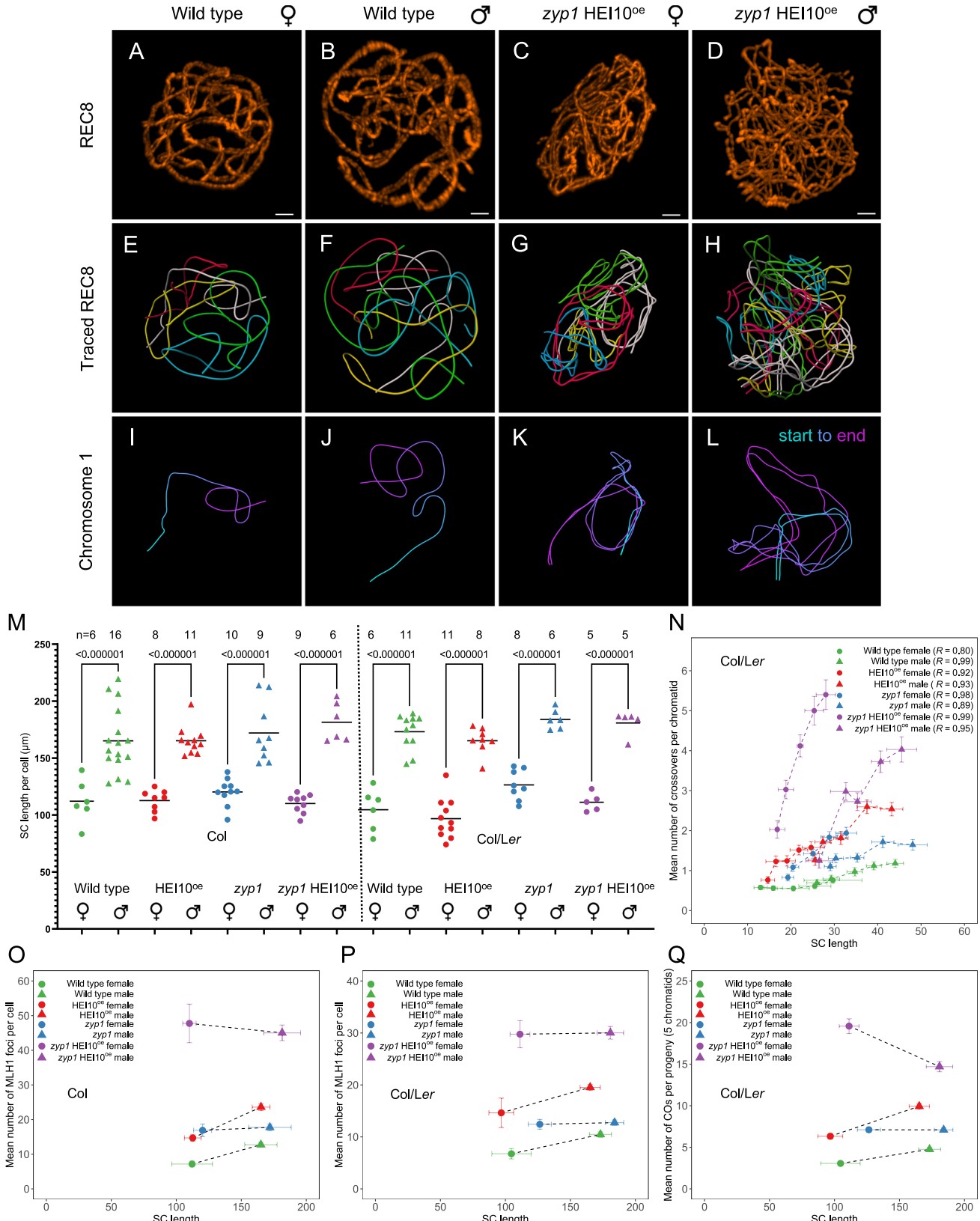

Figure legend shows: Wild type ♀, Wild type ♂, *zyp1* HEI10oe ♀, *zyp1* HEI10oe ♂ with rows labeled REC8, Traced REC8, and Chromosome 1 (start to end).

on the SC depends conjointly on HEI10 expression levels and SC length and that this loading eventually determines CO number.

## The HEI10 coarsening model
The results we present here and previous observations can be interpreted in the context of an emerging model for CO patterning via coarsening through the diffusion of HEI10 along the SC[17,18]. Several models are proposed for CO interference[5,30], but the coarsening model has the advantage of directly accounting for the roles of HEI10 and ZYP1. In this model (Fig. 5), HEI10 initially forms multiple foci along the SC, and HEI10 molecules diffuse along the SC between foci. If larger foci tend to retain more HEI10 molecules than smaller foci, a

**Fig. 4 | Analysis of SC/axis lengths in female and male meiocytes. A–D** REC8 immunolocalization in female and male meiocytes of wild type and *zyp1* HEI10^oe homo (Col). Imaging was done with 3D-STED and the projection is shown. Scale bar = 1 μm. **E–H** REC8 signal was traced in 3D. Each bivalent pair is color-coded. **I–L** Individual trace of the longest chromosome (presumably chromosome 1), with start-to-end color code. **M** Measurement of the total SC length. Each dot is the SC length of an individual cell. Circles and triangles are females and males, respectively. The bars indicate the mean. One-way ANOVA followed by Sidak correction showed that SCs were systematically longer in males than in females ($p < 0.000001$). The same test did not detect any differences between any of the pairs of males of different genotypes ($p > 0.7$). For females, none of the pairwise comparisons were significantly different ($p > 0.13$) except in Col/L*er* HEI10^oe that was lower than Col/L*er zyp1* ($p = 0.006$) and Col *zyp1* ($p = 0.008$). Note that variations in slide preparation and exact meiotic stage may affect this result. The number n of analyzed cells in indicated. **N** Correlation analysis between the mean number of COs per chromosome and SC length (μm) in Col/L*er* background. SCs were attributed to specific chromosomes based on their length (e.g., the longest was presumably chromosome 1). Pearson's correlation coefficients are shown in parentheses. The number of samples in shown in Fig. 4M for SC length and in Fig. 1D for crossovers per chromatid. The relationship between the mean number of MLH1 foci per cell and total SC length per cell in **O** Col background and **P** Col/L*er* background. The number of samples in shown in Fig. 4M for SC length and in Fig. 1B for MLH1 foci per cell. **Q** The relationship between the mean number of COs and SC length in Col/L*er* background. The number of samples in shown in Fig. 4M for SC length and in Fig. 1D for crossovers per chromatid set. The 90% confidence intervals are indicated as error bars. Source data are provided as a Source Data file.

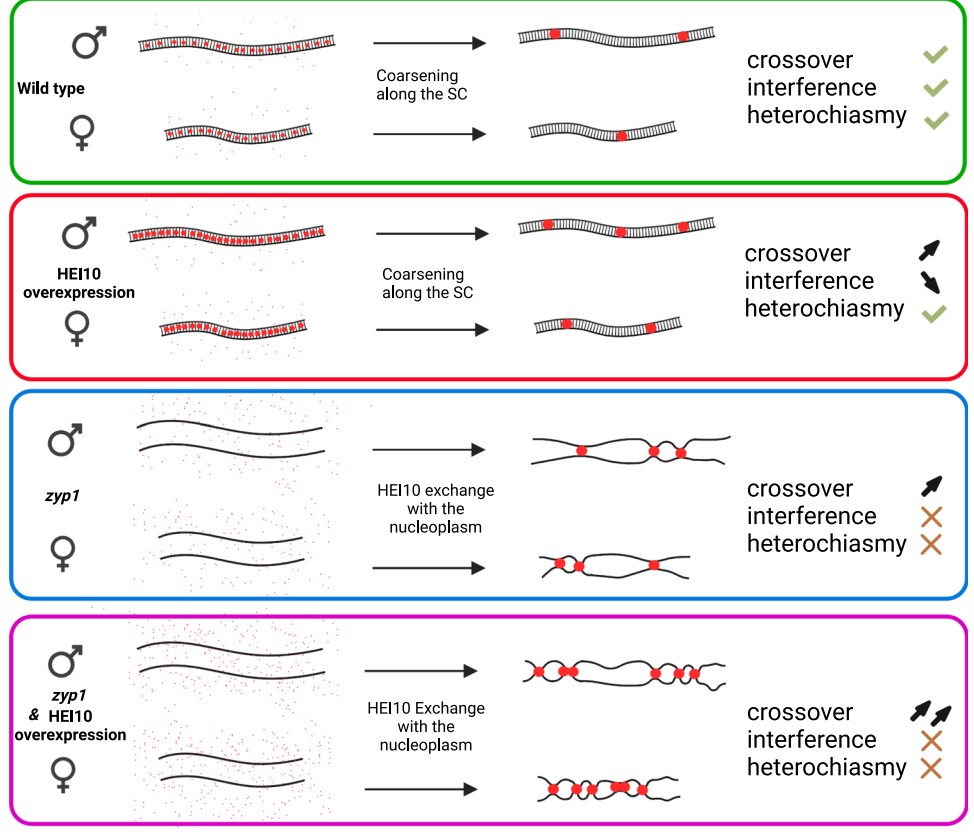

**Fig. 5 | Model of crossover patterning via HEI10 coarsening.** HEI10 (red) is captured at the middle of the SC and coarsens into large pro-CO foci. The number of large pro-CO foci is determined by SC length (heterochiasmy), and HEI10 expression levels. HEI10 overexpression increases CO number, and weakens interference but maintains heterochiasmy. In absence of an SC (*zyp1*), HEI10 is exchanged directly between the foci and the nucleoplasm abolishing both interference and heterochiasmy, and the number of foci depends on HEI10 expression level. Created with BioRender.com.

coarsening process is initiated, and large foci grow at the expense of nearby smaller foci, leading to the formation of well-spaced large foci. These large foci are proposed to create a specific context that promotes class I CO formation (e.g., by attracting the MLH1/MLH3 complex) and protects recombination intermediates from anti-CO factors (i.e., FANCM and RECQ4[24,34]). It is unclear if initial foci colocalize with recombination intermediates or if recombination intermediates favor the coarsening process locally, but both hypotheses envisage final foci to embed such an intermediate. This model predicts the obligate CO, a limited number of COs, and interference[18]. If the coarsening process can proceed without restrictions, it would ultimately lead to the formation of a single focus/CO per bivalent, as observed in *C. elegans*[17]. However, in most species, including Arabidopsis, 2–3 interfering class I COs are typically observed per bivalent. At least three hypotheses can account for this observation: one

proposes an upper limit in the size of foci, above which it stops growing, allowing other foci to be maintained. The second supposes that the coarsening is stopped when a checkpoint is satisfied (e.g., when at least one large focus/CO is formed per chromosome). The third suggests that the process is stopped before completion after a certain period, which we consider here for simplicity. In all cases, the total amount of HEI10 loaded onto the SC determines the number of CO-promoting foci, although in the third case the length of the SC also plays a minor role independently of the total amount of HEI10. The model proposes that two factors jointly determine the initial HEI10 loading: (i) HEI10 concentration in the nucleoplasm, which determines the amount of HEI10 in initial foci and on the SC per μm of SC, and (ii) the length of the SC, which, for a given expression level of HEI10 would determine linearly the total HEI10 loading. Our numerical implementation of this model (see Methods) explains the

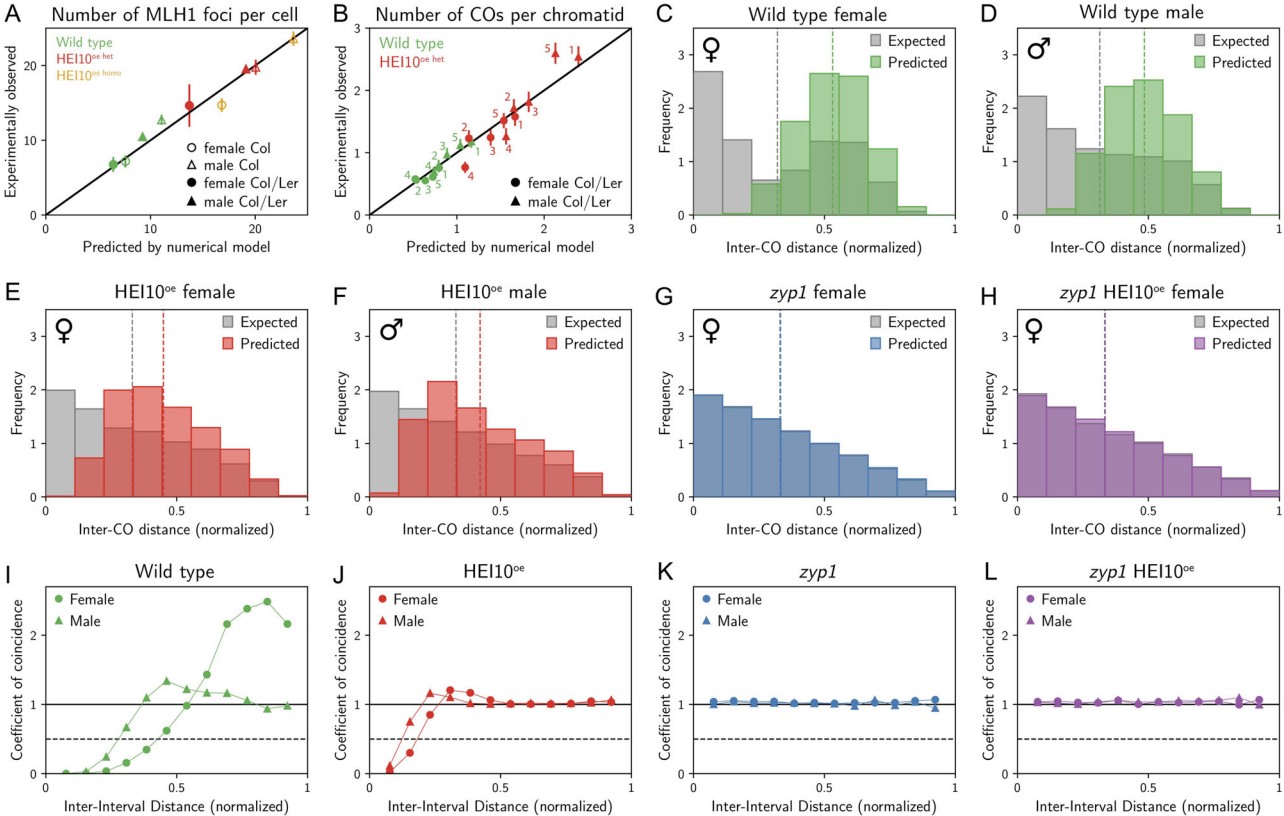

**Fig. 6 | A coarsening model for crossover designation explains the measured data. A** Number of MLH1 foci predicted by the model compared to the experimental measurements shown in Fig. 1B. Error bars denote 90% confidence of the mean. **B** Number of COs per chromatid predicted by the model compared to the experimental measurements shown in Fig. 1D. The respective chromatids are labeled and error bars denote 90% confidence of the mean. **C–H** Predicted distributions of distances between two COs for chromatids with exactly two COs; compare to Fig. 2C–F. Means are indicated by vertical dashed lines. **I–L** Predicted coefficient of coincidence curves; compare to Fig. 2G–J. **A–L** Numerical details are given in the Methods. Numerical predictions were determined from $n = 10000$ (except $n = 1000$ in panel **A**) independent repetitions.

measured CO counts quantitatively (Fig. 6A, B), comparable to ref. 18. In particular, it explains the observed correlation between the length of the SC and the number of COs between chromosome pairs within single cells as well as between different cells, as observed here in Arabidopsis, where female meiosis has a shorter SC and fewer COs than male meiosis. Note that this shorter SC in females also implies stronger CO interference when measured genetically (Fig. 6C, F, I, J; compare with Fig. 2). This model also accounts for the fact that CO number depends on HEI10 expression level, as this level determines the amount of HEI10 loaded per μm of SC[18]. We observed that overexpressing HEI10 increases CO numbers in males and females without eliminating heterochiasmy, as predicted by the difference in SC length and previous modeling[18]. In addition, CO interference is also reduced, but not abolished by over-expressing HEI10, as expected, as the coarsening process still occurs (Fig. 6E, F, J). We propose that in the absence of SC, in the *zyp1* mutant, HEI10 diffusion is no longer constrained to the SC but occurs freely in the nucleoplasm. In this case, foci still form on chromosomes (Fig. 1A, B), but they now exchange HEI10 directly with the nucleoplasm. If this exchange is slow compared to the duration of pachytene, all initial foci grow continuously by taking up HEI10. In contrast, when HEI10 is exchanged more quickly, competition between foci, and thus coarsening, will set in, which was also recently proposed[35]. In both cases, large HEI10 foci form, colocalize with MLH1, and promote class I COs. However, the obligate CO and CO interference are lost as the diffusion is no longer constrained per chromosome (Fig. 6G, H, K, L). In a sense, in the absence of the SC, the coarsening and CO designation process can be said to be "blind" to chromosomes. The absence of the SC must be associated with slower coarsening since otherwise the exchange of HEI10 via the nucleoplasm would be significant in wild type, too. If the number of initial foci in the *zyp1* mutant is roughly comparable to wild type, slower coarsening implies a bigger number of large foci at the end of pachytene, consistent with the increase observed experimentally (Fig. 1). Together with interference, heterochiasmy is abolished when the number of COs per chromosome is solely determined by HEI10 expression level in the nucleoplasm and no longer by HEI10 loading onto the SC. Taken together, the experimental data and the coarsening model show that two factors limit class I COs: ZYP1-mediated CO-interference and HEI10 levels.

A similar model was proposed and further supporting experimental data were recently obtained in *C. elegans*[17,36]. Several additional pieces of evidence suggest that the joint control of COs by SC and HEI10 is conserved: In multiple species, HEI10 homologs also initially form multiple foci before eventually consolidating into a limited number of large foci that co-localize with COs[19–22,37]; COs covary with SC length in many species[13]; Variants that affect recombination rates in natural populations of diverse species involve genes that encode HEI10 homologs[38]. This suggests that the coarsening of HEI10 along the SC may be a conserved process for CO patterning in eukaryotes.

## Methods
### Plant materials and growth conditions
*Arabidopsis thaliana* plants were cultivated in Polyklima growth chambers (16-h day, 21.5 °C, 280 μM; 8-h night, 18 °C: 60% humidity). Wild type Col-0 and L*er*–1 are 186AV1B4 and 213AV1B1 from the Versailles stock center (http://publiclines.versailles.inra.fr/). The *zyp1-1*

(8.7.2V1T3) and *zyp1-6* (1.12V5T2) mutants were characterized previously[8]. The HEI10 over-expression line is Col HEI10 line C2[23], kindly provided by Ian Henderson. Genotyping of the mutants was carried out by PCR amplification (Supplementary Data 2).

To generate the double homozygous mutant *zyp1-1⁻/⁻* HEI10^oe in Col, *zyp1-1⁺/⁻* plants were crossed with HEI10^oe homozygous mutant plants (C2). The obtained double heterozygous *zyp1-1⁺/⁻* HEI10^oe were selfed to produce *zyp1-1⁻/⁻* mutants, HEI10^oe homozygous, and *zyp1-1⁻/⁻* HEI10^oe double homozygous mutants. These sister plants were used to perform MLH1 foci counting, SC measurements, chromosome spreads, and seed countings. To generate *zyp1-1/zyp1-6* HEI10^oe het in Col/L*er*, double heterozygous *zyp1-1⁺/⁻* HEI10^oe (Col) were crossed with *zyp1-6⁺/⁻* (L*er*) to generate *zyp1-1/zyp1-6* HEI10^oe het, HEI10^oe het, *zyp1-1/zyp1-6* and wild type controls in Col/L*er*. These sister plants were used for MLH1 foci counting and SC length measurements and were reciprocally backcrossed with wild type Col to generate the sequencing populations. Backcross populations were grown in the greenhouse for three weeks (16-h day/8-h night) and four days in the dark. For DNA extraction and library preparation, 100–150 mg leaf samples were collected from the four backcross populations[39].

## Cytology

Immunolocalization on male meiocytes were conducted by modifying a previously described method[8,40]. Briefly, fresh 0.35–0.45 mm flower buds were dissected to remove sepals and petals and collected in buffer A (KCl 80 mM, NaCl 20 mM, Pipes-NaOH 15 mM, EGTA 0.5 mM, EDTA 2 mM, Sorbitol 80 mM, DTT 1 mM, Spermine 0.15 mM, spermidine 0.5 mM). Buds were fixed in bufferA+2% formaldehyde for 30 min under vacuum, washed in buffer A for 10 minutes, and digested for 40 minutes at 37 °C (0.3% cellulase, 0.3% pectolyase Y23, 0.3% driselase, 0,1% sodium azide in citrate buffer). Following a wash in buffer A, digested buds were kept in buffer A on ice. Next, 10-15 buds were placed in 6 μL of buffer A on a 18 × 18 mm high precision coverslip, in which anthers were dissected and squashed to extrude meiocytes. A 3 μL drop of activated polyacrylamide solution (25 μL 15% polyacrylamide (SIGMA A3574) in buffer A + 1.25 μL of 20% sodium sulfite + 1.25 μL of 20% ammonium persulfate) is added to the meiocytes and a second coverslip is placed on the top, with gentle pressure. The polyacrylamide gels were left to polymerize for 1 h and then the two coverslips were separated. The coverslips covered by a gel pad were incubated in 1X PBS, 1% Triton X-100, 1 mM EDTA for 1 h with agitation, followed by 2 h incubation in blocking buffer (3% BSA in 1X PBS + 0.1% Tween 20) at room temperature. Coverslips were then incubated with 100 μL of primary antibody in blocking buffer at 4 °C in a humid chamber for 48 h. Coverslips were washed four times 30 min with 1X PBS, 0,1% Triton X-100. One hundred microliters of the appropriate fluorophore-conjugated secondary antibodies in blocking buffer were applied (1:250) and incubated at room temperature for 2 h in the dark. Gels were washed four times 20 min with 1X PBS, 0,1% Triton X-100. 15 μL of SlowFade™ Gold were used for mounting the coverslip with a slide, that was sealed with nail polish.

For female meiocytes, 0.8–1.2 mm pistils were collected and their stigmata cut off. Pistils were then fixed and digested following the same procedure as for male meiocytes, as described above. The pistils were then opened longitudinally and the ovules released on a slide. The subsequent slide treatment and immunolocalization were the same as for male meiocytes.

Four primary antibodies were used: anti-REC8 raised in rat[41] (laboratory code PAK036, dilution 1:250), anti-MLH1 in rabbit[42] (PAK017, 1:200), and anti-HEI10 in chicken[19] (PAK046, 1:5,000). Secondary antibodies (dilution 1:250) were Abberior STAR ORANGE Goat anti-rat IgG (STORANGE-1007), STAR RED Goat anti-chicken IgY (STRED-1005), STAR GREEN Goat anti-rabbit IgG (STGREEN-1002). Super-resolution images were acquired with the Abberior instrument facility line (https://abberior-instruments.com/) 561- and 640-nm

excitation lasers (for STAR Orange and STAR Red, respectively) and a 775-nm STED depletion laser. Confocal images were taken with the same instrument with a 485-nm excitation laser (for STAR GREEN/Alexa488).

## Image processing and analysis

Deconvolution of the images was performed by Huygens Essential (version 21.10, Scientific Volume Imaging, https://svi.nl/) using the classic maximum likelihood estimation algorithm with lateral drift stabilization; signal-to-noise ratio: 7 for STED images and 20 for confocal images, 40 iterations, and quality threshold of 0.5. Maximum intensity projections and contrast adjustments were also done with Huygens Essential 22.04.0p0 64b. Deconvoluted pictures were imported into Imaris x64 9.6.0 (https://imaris.oxinst.com/, Oxford Instruments, UK) for subsequent analysis. MLH1 foci were counted using the spots module in diplotene and diakinesis cells. The vast majority of MLH foci colocalize with a HEI10 focus. Only double MLH1/HEI10 foci present on chromosomes were taken into account. For REC8 signal tracing, fully synapsed cells were used to trace the chromosomes. In wild type and HEI10^oe, the five synapsed bivalents were traced. In *zyp1* and *zyp1* HEI10^oe, five pairs of parallel chromosomes were traced. The surface module was used to create a clean masked REC8 channel for filament tracing. The filament module was used to trace the SC length, AutoDepth function was used to do semi-automatic tracing and get the simulated chromosome. The SC length of each chromosome was measured using the statistics function of the Filament module. Statistical tests were performed in Prism 9.4.1.

## CO identification and analysis

In this study, the female and male population of wild type (48 and 47 plants), HEI10^oe (144 and 141 plants), *zyp1* (48 and 47 plants) and *zyp1* HEI10^oe (142 and 138 plants) were sequenced by Illumina HiSeq3000 (2 × 150bp) conducted by the Max Planck-Genome-center (https://mpgc.mpipz.mpg.de/home/). The raw sequencing data of the female and male population of wild type (212 and 120 plants, respectively) and *zyp1* (224 and 178 plants) from a previous study (ArrayExpress number E-MTAB-9593)[8] were also included in this study. In total, we analyzed 260 and 167 wild type female and male, 144 and 141 HEI10^oe female and male, 272 and 225 *zyp1* female and male, 142 and 138 *zyp1* HEI10^oe female and male plants, separately. The raw sequencing data were quality-controlled using FastQC v0.11.9 (http://www.bioinformatics.babraham.ac.uk/projects/fastqc/). The sequencing reads were aligned to the *Arabidopsis thaliana* Col-0 TAIR10 reference genome, which was downloaded from TAIR[43,44] (https://www.arabidopsis.org/), using BWA v0.7.15-r1140[45], with default parameters. A set of Sambamba v0.6.8[46] commands was used for sorting and removing duplicated mapped reads. Whole-genome alignment between the Col TAIR10 reference genome and the L*er* assembly[25] (https://1001genomes.org/data/MPIPZ/MPIPZJiao2020/releases/current/strains/Ler/) was performed by nucmer from the MUMmer4 toolbox[47], with parameters "–maxmatch -c 100 -b 500 -l 50", and further filtered by delta-filter, with parameters "-m -i 90 -l 100". Then, SyRI v1.2[48] was used to identify structural rearrangements and local variations (SNPs and small indels); the identified syntenic SNPs were used for the generation of a high-confidence SNP marker list, and the detected inversions and translocations (>100 bp) were used for the chromosomal comparison with CO distribution (Supplementary Fig. 5). To call CO we used a set of 620,115 high-confident SNP markers and a sliding window-based method, with a window size of 50 kb and a step size of 25 kb[8]. Samples with low coverage (<0.1× depth) or a potential contamination (more than 5% of the windows with the Col allele frequency in the range of 0.8–0.9) were filtered out[8,31,49–51]. Samples of each population were randomly selected for checking predicted COs manually by inGAP-family[50]. To simulate the distribution of CO distances in the absence of CO interference we: (i) For each chromosome, computed the distances between all

possible pairs of CO positions observed in the entire population of chromosomes with exactly two COs (e.g., distance from a CO in sample 1 with a CO in sample 2). (ii) To match the relative contribution of each chromosome between the observed dataset and the simulation, we downsized the simulated dataset of individual chromosome by random sampling. We used a total of 1608, 14,208, 23,628, 26,320, 57,000, 41,548, 4704, and 9944 simulated chromosomes for female/male wild type, HEI10[oe], *zyp1*, and *zyp1* HEI10[oe], respectively. (iii) We calculated the distribution of CO distances as for the observed data. The Coefficient of Coincidence (CoC) was calculated for CO interference analysis using MADpattern[30,52], with a number of 13 intervals. Chromosome 4 was excluded from interference analyses because of a translocation associated with the HEI10 transgene[29] and potential inversion in our L*er* line[8]. To profile the CO distribution along chromosomes, CO position was defined randomly in the range of CO interval and a sliding window-based strategy was used, with 1 Mb window size and 50 kb step size. Then, the local distribution of recombination (CO resolution ≤1000 bp) was explored by ChIPseeker v1.22.1[53], with the promoter region defined as 2000 bp upstream of the transcription start site.

### Aneuploidy screening by whole-genome sequencing

The sequencing depth of each non-overlapping 100 kb window across the genome was evaluated by Mosdepth v0.2.7[54] with parameters of "-n-fast-mode-by 10000". For each sample, pairwise testing of sequencing depths along chromosomes was performed using the Mann–Whitney test, and significant $p$ values were adjusted using the fdr method. A pair of tested chromosomes with fold change >1.2 and $p$ value <1e−20 was considered as aneuploid.

### Mathematical model of CO patterning

The mathematical model we use is equivalent to the one presented in ref. 18. To account for diffusion of HEI10 along the SC and the exchange of HEI10 between SC and foci, it describes the concentration $c(x, t)$ of HEI10 along the SC of length $L$ together with the amounts $M_i(t)$ of HEI10 in $N$ foci that are placed at positions $x_i$ along the SC for $i = 1, \ldots, N$. Here, $x$ denotes the position along the SC and $t$ denotes time. Focus $i$ grows if the local HEI10 concentration on the SC, $c(x_i)$, is larger than the equilibrium concentration

$$c^{\mathrm{eq}}(M_i) = c_0^{\mathrm{eq}} \frac{M_i}{1 + M_i^{1+\alpha}}, \tag{1}$$

implying the growth dynamics

$$\frac{\mathrm{d}M_i}{\mathrm{d}t} = \Lambda\left[c(x_i) - c^{\mathrm{eq}}(M_i)\right], \tag{2}$$

where $\Lambda$ quantifies the rate of HEI10 exchange. HEI10 diffuses with diffusivity $D$ along the SC and is exchanged with foci,

$$\frac{\partial c}{\partial t} = D\frac{\partial^2 c}{\partial x^2} - \Lambda \sum_{i=1}^{N} \delta(x - x_i)\left[c(x_i) - c^{\mathrm{eq}}(M_i)\right]. \tag{3}$$

We impose no-flux boundary conditions at $x = 0$ and $x = L$, so the total amount of HEI10 is conserved. We implemented this model using finite differences by discretizing the SC using 50 grid points and solved the resulting equations using an explicit Euler scheme[55]. We initialize the system with a uniform concentration on the SC, $c(x, t = 0) = c_{\mathrm{init}}$. The $N$ foci are positioned uniformly along the SC and their sizes $M_i$ are chosen independently from a normal distribution with mean $M_{\mathrm{init}}$ and standard deviation $\sigma$, which has been truncated to $[M_{\mathrm{init}} - 3\sigma, M_{\mathrm{init}} + 3\sigma]$. The diffusivity $D = 1.1\,\mu\mathrm{m}^2/\mathrm{s}$, the exchange rate $\Lambda = 2.1\,\mu\mathrm{m/s}$, the exponent $\alpha = 0.25$, and the base equilibrium concentration $c_0^{\mathrm{eq}} = 1.35$ a.u./$\mu$m, are inspired by ref. 18. We use SC

lengths $L$ measured in wild type (Fig. 4N) and estimate an initial density of four foci per $\mu$m, based on cytology[8]. For simulations, we choose $M_{\mathrm{init}} = y \cdot 3.4$ a.u. , $\sigma_{\mathrm{init}} = y \cdot 1.1$ a.u., and $c_{\mathrm{init}} = y \cdot 1.4$ a.u./$\mu$m, where $y$ is a factor to account for higher HEI10 expression levels. We chose $y = 2$ for wild type Col, $y = 6$ for HEI10[oe het] Col, $y = 8$ for HEI10[oe homo] Col, $y = 1.5$ for wild type Col/L*er*, and $y = 5.5$ for HEI10[oe het] Col/L*er*, which accounts for the reduced activity in L*er*[56] and HEI10 overexpression. We simulate coarsening on each individual SC for male and female meiosis for 10h, comparable to the duration of pachytene[57,58]. Only foci above a threshold size of $M_{\mathrm{thresh}} = 3$ a.u. are assumed to attract MLH1 and form class I COs. The associated COs per chromatid were determine by choosing COs from the bivalent independently with 50% probability. Here, we assume a linear relation between the position of the MLH1 foci on the SC ($\mu$m) and the genomic position (Mb) of the COs on the chromatid. Note that we excluded chromosome 4 from the interference analyses, consistent with the analysis of the experimental data described above.

In the case of the *zyp1* mutant, the model implies that all CO positions are independent. To obtain a theoretical distribution of COs, we thus first determine the number of COs per chromatid by sampling a Poisson distribution with a mean given by the experimental data (Fig. 4N) and then distribute these COs uniformly along the chromatid length.

### Reporting summary

Further information on research design is available in the Nature Research Reporting Summary linked to this article.

### Data availability

The list of identified COs in the female and male populations of wild type, HEI10[oe], *zyp1*, and *zyp1* HEI10[oe] can be accessed in Supplementary Data 1. The raw sequencing data generated in this study have been deposited in the ArrayExpress EMBL-EBI database under accession code E-MTAB-11696. The reference Col (TAIR10) and L*er* genomes used in this study can be found in https://www.arabidopsis.org/ and https://1001genomes.org/data/MPIPZ/MPIPZJiao2020/releases/current/strains/Ler/, respectively. Source data are provided with this paper.

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

## Acknowledgements

This work was supported by core funding from the Max Planck Society to R.M., M.E., as well as D.Z. and Alexander von Humboldt Fellowships to Q.L. and J.J. The IJPB benefits from the support of Saclay Plant Sciences-SPS (ANR-17-EUR-0007). We thank Abby Dernburg for enlightening discussions. We thank Ian Henderson for kindly providing the C2 line. We thank Neysan Donnelly for proofreading the manuscript.

## Author contributions

S.D. produced and analyzed the MLH1-HEI10 cytological data, developed the protocol for female immunolocalization, produced all the genetic material, and analyzed fertility. Q.L. analyzed the sequencing data and performed recombination, interference, and aneuploidy analyses. J.J. generated SC images, and analyzed chromosome mis-segregation and SC length data. M.E. and D.Z. developed and analyzed the mathematical model of CO patterning. M.G. developed the method for chromosome 3D analyses. R.M. lead the project and wrote the manuscript with input from all co-authors.

## Funding

## Competing interests

The authors declare no competing interests.
