## [Peer Review File · Nature Communications]

Joint control of meiotic crossover patterning by the synaptonemal complex and HEI10 dosageReviewers' Comments:

Reviewer #1:

Remarks to the Author:

During meiosis, the number and distribution of crossovers (COs) are tightly controlled. However, the mechanistic basis of this control is still debated. In most eukaryotes, each chromosomal bivalent must receive at least one CO (the obligate CO) and COs on the same bivalent tend to be distantly spaced (CO interference). Here, Durand et al., use a combination of genetics, cytology and mathematical modelling to investigate the combined effects of HEI10 overexpression and synaptonemal complex loss on meiotic CO frequency and the genetic recombination landscape in Arabidopsis.

The authors have generated a very impressive amount of data in this study (including male and female cytology and NGS data from a large number of mutants and crosses). Importantly, they demonstrate that combined HEI10 overexpression in a *zyp1* mutant background leads to a massive increase in class I COs (building on previous observations from the Mercier lab, and others, that *zyp1* null mutants and HEI10 overexpressors, individually, increase class I COs to a smaller extent). Intriguingly, they also find that female and male axis lengths are unaffected by HEI10 overexpression or *zyp1* mutation. These are both novel and interesting discoveries that will be of great interest to the wider meiosis community.

However, there are a number of major and minor issues with the paper that the authors should address:

Major Points:

1. I have serious reservations about the section titled "High CO rates in *zyp1* and HEI10oe are not associated with meiotic defects". Firstly, this statement is clearly incorrect as the authors demonstrate there are, in fact, some meiotic defects in these lines evidenced by metaphase univalents, aneuploid offspring and reduced fertility. Additionally, much of the analysis is poorly explained and interpreted. The authors use sequencing depth analysis to identify trisomic plants in their population. However, they refer to analysis of aneuploid "gametes" in the text, which is not at all what they have sequenced. They have sequenced the product of a gamete that has accomplished successful fertilisation with another gamete to generate a viable and (presumably) healthy plant. There are a whole host of reasons why the number of aneuploid plants detected would be significantly fewer than the number of aneuploid gametes (e.g. low transmission of trisomic gametes and poor viability/germination of aneuploid plants with different karyotypes). For example, it is known that transmission of gametes with an extra chromosome is low in Arabidopsis (particularly male transmission, with the exception of chromosome 4. See Method in Arabidopsis research, ISBN: 9810209045). This might explain why only chromosome 4 trisomics are detected. As the authors also know, the absence of a genetically detectable CO does not mean that a CO did not occur on the other sister chromatid during meiosis and, likewise, that the genetic detection of 2 COs does not exclude the probable likelihood that additional COs occurred on the other chromatids. The authors cannot, therefore, conclude that "the aneuploidies thus appear to be associated with the absence of COs or specific configurations of a pair of COs". It is also interesting that aneuploid offspring were detected from the HEI10oe C2 plants, which presumably maintain CO assurance. Could this be due to the presence of a translocation that was previously reported in the HEI10oe C2 line (doi.org/10.1073/pnas.1713071115)? Do the authors know if their material still contains this translocation? If so, this should be commented upon.

2. The interpretation and discussion of CO interference within the manuscript needs improving. As noted by the authors (lines 61-64) the relevant metric for measuring CO interference is axis/SC length. A 'Matters Arising' was also recently published in Nature that highlights the limitations of using genetic sequencing data to understand crossover patterning (doi.org/10.1038/s41586-022-04693-2). Despite this, all inferences relating to interference within this manuscript are based on the analysis of

genetic CO data. The drawbacks of this approach should therefore be made clear within the manuscript. The manuscript would also benefit from a more thorough explanation of why CoC analysis is a "likely more accurate" measure of interference (line 174), rather than histograms or CDF plots of inter-CO distances? I'm also curious as to why 13 intervals were specifically chosen for the CoC analysis? Does the analysis produce comparable results for a greater or fewer number of intervals?

3. The title of the article is confusing. Firstly, it is impossible to tell what the article is about from this title. Secondly, are the authors not making the argument that the SC and HEI10 dosage are just two component parts of a singular (not "dual") control mechanism for meiotic crossover patterning (i.e., coarsening)? The title therefore needs to be changed.

4. I think figure 5 is a useful addition to the paper. However, it needs to be modified to more accurately represent the mathematical coarsening model described in the paper. In the HEI10 overexpression panel there are a greater number of small foci than in the WT. However, if these small foci are located at recombination intermediate sites then we would not expect the number of small foci to change compared with WT, but merely their initial size will be bigger (and also there will be more background HEI10 along the SC). In the *zyp1* null mutants, similarly, we would still expect to initially see some small foci between the partially co-aligned axes, which then coarsen down to a few (randomly distributed) larger foci.

5. It needs to be made much clearer that the model the authors use is actually that from Ref. 6 <https://doi.org/10.1038/s41467-021-24827-w>, with only minor modifications in the parameters chosen. For example, in line 408, it is described as "our mathematical model", in line 410 "Similar to the model presented in ref. 6", in line 426 "inspired by ref. 6". In all these cases the model is actually the same! If this is not spelled out with much greater precision, the field will end up being completely confused as to which model is being simulated and how various different models are different (or not) to each other. In addition, the way the model is treated is not well integrated into the rest of the manuscript, with the modelling section seemingly bolted on to the end of the manuscript.

6. The authors use the terms "droplets" and "condensates" to describe HEI10 behaviour throughout the paper. "Foci" is more appropriate as the liquid-like properties of HEI10 clusters have yet to be fully experimentally demonstrated.

Minor points:

7. Figure 1. Could the authors also include a representative image which shows HEI10 staining (perhaps as a supplementary figure)?

8. Figure 1. Figure 1 E, x-axis should be "Mb", not "M".

9. Figure 2 legend. Could they explain how the "permutation analysis" was performed in the methods?

10. Lines 169-170. It should be mentioned that CO interference has already been measured in HEI10oe C2 lines both genetically and cytologically (<https://doi.org/10.1073/pnas.1713071115>, <https://doi.org/10.1038/s41467-021-24827-w>).

11. Lines 289-299. It needs to be mentioned that the effects of HEI10 overexpression, underexpression and heterochiasmy on coarsening have already been modelled in *Arabidopsis* (<https://doi.org/10.1038/s41467-021-24827-w>).

12. Line 293: "shorter SC length in females implies stronger crossover interference". It should be made clear that this might only apply when interference is measured genetically (see point 2).

13. Figure 6A. For the measured number of MLH1 foci per cell, why is some experimental data (from

Fig 1B) missing or different? For example, male/female Col HEI10oe homo is missing, but female Col HEI10oe het is present on this plot but not in Fig 1B).

14. The authors often refer to the "central element" of the SC (e.g., lines 29, 54) and its role in interference. This is a bit confusing as the central element is a specific structural feature of the SC that runs along the centre of the SC (where the N-termini of the transverse filaments overlap) and the central element of the SC in Arabidopsis has not yet been characterised. The authors should therefore just refer to the "SC" more generally, rather than specifically to the "central element of the SC".

15. Line 126. As well as mentioning that HEI10oe lines maintain heterochiasmy, but *zyp1* mutants do not, they could also mention that HEI10oe lines maintain CO interference (albeit weaker) but *zyp1* mutants do not.

16. Lines 154-155. They should make it clear that the "physical size" of the chromosomes refers to the genomic size, measured in Mb.

17. Line 228. "Female and male SC lengths differ" should be changed to "Female and male axis lengths differ", as obviously there is no SC in the *zyp1* mutant.

18. Line 396. Why was chromosome 4 excluded from interference analysis?

19. Figure S5. The colour of the points for Male and Female COs are the same in all four plots. Is it supposed to be this way (in which case, why have they provided the key)? Is there any way to produce a plot with the male and female COs in different colours?

Reviewer #2:

Remarks to the Author:

The paper of Durand et al. builds on earlier works showing the key role of HEI10 in limiting crossover recombination (Ziolkowski et al. 2017) and SC in the control of crossover interference, crossover assurance and heterochiasmy (Capilla-Pérez et al. 2021; France et al. 2021) in Arabidopsis. The authors show that HEI10 overexpression leads to a significant reduction in interference, as has been shown to some extent before, but for the first time they show that HEI10 overexpressing lines maintain heterochiasmy. In contrast, crossover interference and heterochiasmy completely disappear in lines with *zyp1* mutation, a key SC element. The authors used a number of demanding cytological and genetic techniques that allowed them to obtain a comprehensive picture of the recombination events taking place in the analyzed lines. By combining overexpression of the HEI10 protein with a mutation in the *ZYP1* gene the authors showed that it is possible to significantly break the limit for the number of class I crossovers (possibly also to some extent class II) with 20 crossovers per gamete (and above 40 COs in some cases), with a complete loss of interference, assurance and heterochiasmy.

The most general conclusion is reflected in the title, saying that both HEI10 and *ZYP1* contribute to the control of meiotic crossover frequency and patterning in Arabidopsis. Moreover, the presented results constitute an important test for the recombination nodule/HEI10 coarsening model (Zhang et al. 2021; Morgan et al. 2021), which is now widely discussed in the field, providing a further support for this model. Thus, this work is of great importance to our understanding of crossover interference. Overall, I believe this is a work that significantly advances our understanding of meiosis and crossover recombination and would be of great interest for a broad audience of Nature Commun. readers. Nevertheless, one of the main results of this work needs to be experimentally validated to be sure that the conclusions drawn are correct (see below, point 1).

Major points:

1.

The authors observed that crossover number in female *zyp1* HEI10-oe inferred from BC1 sequencing is significantly higher than the number of MLH1 foci and hypothesized that this is due to an increase in Class II crossovers. While I consider such an explanation as the most probable, it raises another serious problem that is key to some of important conclusions of this publication: If female *zyp1* HEI10-oe experiences increased class II crossovers activity, this effect may be due either to HEI10 overexpression, or the *zyp1* mutation. The first of these possibilities does not seem probable, because the authors do not see a similar effect in crosses with C2 line (not carrying the *zyp1* mutation). Therefore, one can conclude that the loss of heterochiasmy observed in *zyp1* is apparent and results in fact from hyperactivation of class II crossovers in the female *zyp1* line. The consequence of this could be a specific "opposite" heterochiasmy in the *zyp1* HEI10-oe line, with a higher crossover number in female than in male meiosis.

There are many speculations which could be drawn from this explanation: This would suggest that a difference in class II crossover activity between male and female meiosis contributes to heterochiasmy, which would be equally interesting. Moreover, this would indicate that HEI10 stimulates not only ZMM pathway but also class II crossovers (which I think fits well to its predicted role in the coarsening model, i.e., protection of intermediates from DNA helicases). Perhaps it would even shift the responsibility for heterochiasmy/assurance?/interference? from SC to the chromosome axis.

Anyway, the authors need to carry out another BC1 sequencing experiment, this time for the *mus81* mutant (in a combination of *mus81 zyp1* and/or *mus81 zyp1* HEI10-oe; both female and male meiosis). I understand that it can be challenging as an *mus81* mutant needs to be created in Ler background, but I think this is required to support the conclusion about the loss of heterochiasmy in *zyp1* and *zyp1* HEI10-oe and the role of SC in crossover sex dimorphism. I also believe that this would provide us a much deeper understanding of crossover control.

2.

Wild type Col inbreds are significantly hotter than Col/Ler hybrids, which was explained in the paper by a more effective pro-crossover HEI10 allele in Col accession. In theory, the same could apply to a difference between *zyp1* inbreds and hybrids. However, even a more dramatic difference is observed between *zyp1* HEI10-oe inbreds and hybrids. It seems unlikely that this would result from abovementioned HEI10 allele activity, since the HEI10 overexpression is close to saturation (line 107). Can the authors propose why Col/Ler hybrids show lower CO in this genetic context than Col inbreds?

I think that the results presented on Fig. 4 exclude the possibility that this is linked to differences in chromosome axis length between inbreds and hybrids (as there are no differences). Is it possible that in the context of the absence of SC and the limited contact between homologous chromosomes (pairing but no synapsis), polymorphism between homologues restricts recombination to a greater extent than it does in WT, when SC is fully functional? To verify this hypothesis, the authors should conduct a more in-depth analysis of CO distribution against the polymorphism pattern along the chromosomes. The method of carrying out such an analysis is left to the authors' discretion.

Judging from the CO distribution maps (especially in female meiosis) in Figs. 2A and S3, certain chromosomal regions show much greater crossover suppression in *zyp1* HEI10-oe than WT, e.g. regions around 22 and 25 Mb on chromosome 1. Could the authors investigate what is so special in these regions?

Of course, other explanations for inbred/hybrid difference are also possible including different activity of Class II crossover pathways in Col and Ler accessions, and this again would be tested with the experiment proposed by me in point 1.

3.

I like the part about the HEI10 coarsening model which I think is very clear and easy to understand. However, in Fig. 6 I miss the application of the model to the combination of *zyp1* HEI10-oe. Can the authors complete the figure with this genetic background?

Minor comment:

- Fig. 1B and 1D I think it would be useful to show on the plot the statistical significance between male and female meiosis for each genotype.

Reviewer #3:

Remarks to the Author:

The data presented by Durand et al builds on recent publications in Arabidopsis that crossover interference is abolished in the *zyp1* mutant and that over-expressing HEI10 increases crossovers. It is of very high quality and of great interest to the meiosis field. The main finding is that combining the *zyp1* mutants and HEI10 overexpressing lines increase class I crossovers in Arabidopsis, determined by cytological markers such as MLH1/HEI10 and sequencing products from *col/ler* crosses. This is different to the publications on individual genes as it indicates that there is a finite pool of HEI10 protein that limits the numbers of crossovers in wild type and that this protein needs to be distributed to all chromosome pairs to maintain the obligate crossover. The authors consider that the HEI10 coarsening model explains the observed data, but it would be useful for the reader if the authors explain why other models are discounted. For example, why is that ZYP1 does not prevent closely spaced HEI10 foci and why is PCH2 not important in CO interference as suggested by Yang et al 'Bipartite recruitment of PCH2....'?

Points to be corrected

Lines 14 and 15 are badly written and should be improved, especially as it is these sentences start this section.

Line 20. I know what you mean, but where is the 'H' for HEI10 in ZMM in proteins? This could be confusing for the non-expert. HEI10 as a class I CO protein is a better description.

Line 21. 'shorter axis lengths' is a better way to write that statement

Line 38. This is confusing because the previous sentence has mentioned the ZMM proteins and class I COs, but there is not a vast excess of DSBs in yeast compared to COs. It's considered to be about double. Therefore, this sentence requires more precision and a reference.

Line 68. encodes 'a' not 'an'.

Line 71. Transverse filament, not transverse element

Line 89. Sometimes you use 'CO', but sometimes 'crossover', so consistency would be good throughout this manuscript.

Line 93. It's not common to refer to genotypes in the plural, so wild type not wild types is better.

Line 115. Transverse filament!

Line 139. I don't follow this. If there are 14.7 CO per gamete and there are 4 gametes per meiosis then the number of MLH1 foci should be 14.7×4 at prophase, so why is it 30?

Line 144. I also don't follow the logic of how you can infer that class II COs are increased by this analysis. Please explain.

Line 146. For consistency please use 'transverse filament' for ZYP1 as it is not a central element protein per se, although the central element does form in this mutant.

Line 160. You've performed an 'assay', not written an 'essay'.

Line 221. Ensure, not insure

Line 231. Axes not axis

Line 318. Evidence not evidences

References need correcting

Reviewer #1 (Remarks to the Author):

During meiosis, the number and distribution of crossovers (COs) are tightly controlled. However, the mechanistic basis of this control is still debated. In most eukaryotes, each chromosomal bivalent must receive at least one CO (the obligate CO) and COs on the same bivalent tend to be distantly spaced (CO interference). Here, Durand et al., use a combination of genetics, cytology and mathematical modelling to investigate the combined effects of HEI10 overexpression and synaptonemal complex loss on meiotic CO frequency and the genetic recombination landscape in Arabidopsis.

*The authors have generated a very impressive amount of data in this study (including male and female cytology and NGS data from a large number of mutants and crosses). Importantly, they demonstrate that combined HEI10 overexpression in a *zyp1* mutant background leads to a massive increase in class I COs (building on previous observations from the Mercier lab, and others, that *zyp1* null mutants and HEI10 overexpressors, individually, increase class I COs to a smaller extent). Intriguingly, they also find that female and male axis lengths are unaffected by HEI10 overexpression or *zyp1* mutation. These are both novel and interesting discoveries that will be of great interest to the wider meiosis community.*

> We are pleased that the reviewer appreciates the work.

However, there are a number of major and minor issues with the paper that the authors should address:

Major Points:

*1. I have serious reservations about the section titled “High CO rates in *zyp1* and HEI10oe are not associated with meiotic defects”. Firstly, this statement is clearly incorrect as the authors demonstrate there are, in fact, some meiotic defects in these lines evidenced by metaphase univalents, aneuploid offspring and reduced fertility.*

> Thank you for pointing this out. We were trying to emphasize that the mild meiotic defect that we observed is associated with the events with low numbers of COs rather than with high CO numbers. The few cases of aneuploidies were associated mostly with 0 COs on the transmitted chromatid, while the number of COs per chromatid on the same chromosome is on average 2. Most notably, the extreme cases with up to seven COs on one chromatid of chromosome 4 are not associated with aneuploidy. We have rewritten this section to make it clearer that meiotic defects are indeed present and edited the title of the section.

Additionally, much of the analysis is poorly explained and interpreted. The authors use sequencing depth analysis to identify trisomic plants in their population. However, they refer to analysis of aneuploid “gametes” in the text, which is not at all what they have sequenced. They have sequenced the product of a gamete that has accomplished successful fertilisation with another gamete to generate a viable and (presumably) healthy plant. There are a whole host of reasons why the number of aneuploid plants detected would be significantly fewer than the number of aneuploid gametes (e.g. low transmission of trisomic gametes and poor viability/germination of aneuploid plants with different karyotypes). For example, it is known that transmission of gametes with an extra chromosome is low in Arabidopsis (particularly male transmission, with the exception of chromosome 4. See Method in Arabidopsis research, ISBN: 9810209045). This might explain why only chromosome 4 trisomics are detected.

> We used gametes for simplicity but completely agree with the point made by the reviewer. We use “transmitted chromatid” instead of “gamete” in the revised manuscript. We also discuss the limitation of the analysis and notably the fact that aneuploids are counter-selected and that missegregation is consequently underestimated.

As the authors also know, the absence of a genetically detectable CO does not mean that a CO did not occur on the other sister chromatid during meiosis and, likewise, that the genetic detection of 2 COs does not exclude the probable likelihood that additional COs occurred on the other chromatids. The authors cannot, therefore, conclude that “the aneuploidies thus appear to be associated with the absence of COs or specific configurations of a pair of COs”. It is also interesting that aneuploid offspring were detected from the HEI10oe C2 plants, which presumably maintain CO assurance. Could this be due to the presence of a translocation that was previously reported in the HEI10oe C2 line (doi.org/10.1073/pnas.1713071115)? Do the authors know if their material still contains this translocation? If so, this should be commented upon.

> Yes, indeed. Our point was that the aneuploid chromatid appears to be in the lower range of the distribution of COs per transmitted chromatid (from 0 to 7 per chromatid for chromosome 4 (New Figure S3). It is intriguing that the aneuploids are not associated with the chromatid with the highest number of COs. We think that these observations support – but do not prove, as rightly pointed out by the reviewer – the hypothesis that the absence of COs is one of the causes of aneuploidy. We have rewritten the section accordingly. For the C2 plants, the translocation is indeed still present, as it is associated directly with the HE10 transgene. We have made this clear in the revised manuscript.

2. The interpretation and discussion of CO interference within the manuscript needs improving. As noted by the authors (lines 61-64) the relevant metric for measuring CO interference is axis/SC length. A ‘Matters Arising’ was also recently published in Nature that highlights the limitations of using genetic sequencing data to understand crossover patterning (doi.org/10.1038/s41586-022-04693-2). Despite this, all inferences relating to interference within this manuscript are based on the analysis of genetic CO data. The drawbacks of this approach should therefore be made clear within the manuscript. The manuscript would also benefit from a more thorough explanation of why CoC analysis is a “likely more accurate” measure of interference (line 174), rather than histograms or CDF plots of inter-CO distances? I’m also curious as to why 13 intervals were specifically chosen for the CoC analysis? Does the analysis produce comparable results for a greater or fewer number of intervals?

> The relevant metric for CO interference is a matter of debate. It is clear that axis/SC length is the relevant space for the mechanism of interference. Whatever the nature of the interference signal, there is a lot of evidence that it propagates in the axis/SC space rather than in the genomic space (Mb space). There is thus no debate that the space in which CO distribution is determined is the axis (μm). However, in terms of genetic consequences (i.e., the low frequency of certain allele combinations), the DNA/genic space is relevant. For example, it is likely that the determinants of interference are the same in male and female Arabidopsis (only the axis/SC length is different), but it appears to us that saying that interference is identical in males and females is inaccurate (the mechanism is the same, but the genetic consequences are different). We believe that we actually fully agree with the reviewer, and that this is just a matter of being clear in the definitions. We have edited the manuscript to avoid ambiguities and made clear the limitation associated with the use of genetic data rather than cytological data.

The major limitation of using inter-CO distances is that inter-CO distances are affected by other parameters, and most notably by the number of COs (The crowding effect mentioned in the “Matters

arising” by Kleckner and colleagues) (by the way, this applies to both μm and Mb spaces). This measure is still useful as it gives a very intuitive way to detect the presence of interference. However, using it to compare interference in different conditions may be inaccurate as phenomena other than interference, including the crowding effect, may modify the distribution (discussed in [10.1016/j.semcd.2016.02.024](https://doi.org/10.1016/j.semcd.2016.02.024) and [10.1371/journal.pgen.1004042](https://doi.org/10.1371/journal.pgen.1004042)). The coefficient of coincidence is by construction independent of the frequency of COs and is thus a more direct measure of interference. One nice illustration of this is that if one randomly eliminates half of the COs in an experimental or simulated dataset (and thus reduces the number COs by a factor of 2), the CoC curves remain identical. This is particularly relevant when analyzing genetic data. In addition, CoC analysis uses all the data available, while inter CO distances use only a subset of the data (notably ignoring SC or gametes with a single CO). Finally, the CoC curves hold the advantage of allowing easy assessment of the distance at which the inhibitory effect vanishes.

Thirteen intervals is a compromise between having the largest number of points, and the sampling noise associated with undersized intervals. Very comparable results are obtained with more or fewer intervals, as shown in the novel Figure S8.

3. The title of the article is confusing. Firstly, it is impossible to tell what the article is about from this title. Secondly, are the authors not making the argument that the SC and HEI10 dosage are just two component parts of a singular (not “dual”) control mechanism for meiotic crossover patterning (i.e., coarsening)? The title therefore needs to be changed.

> We changed the title to “Joint control of meiotic crossover patterning by the synaptonemal complex and HEI10 dosage”

*4. I think figure 5 is a useful addition to the paper. However, it needs to be modified to more accurately represent the mathematical coarsening model described in the paper. In the HEI10 overexpression panel there are a greater number of small foci than in the WT. However, if these small foci are located at recombination intermediate sites then we would not expect the number of small foci to change compared with WT, but merely their initial size will be bigger (and also there will be more background HEI10 along the SC). In the *zyp1* null mutants, similarly, we would still expect to initially see some small foci between the partially co-aligned axes, which then coarsen down to a few (randomly distributed) larger foci.*

> We are unsure if HEI10 overexpression leads to more numerous and/or bigger initial foci. It is also unclear if the initial foci co-localize with the recombination sites, or if recombination sites favor HEI10 coarsening (as discussed in the manuscript). In the modeling, more or bigger initial foci lead to a similar outcome. We thus modified Figure 5 to include a mix of bigger and more numerous foci. In the *zyp1* background, we do not see the equivalent of the well-defined initial foci that form along the SC in wild type, but rather a diffuse signal.

5. It needs to be made much clearer that the model the authors use is actually that from Ref. 6 <https://doi.org/10.1038/s41467-021-24827-w>, with only minor modifications in the parameters chosen. For example, in line 408, it is described as “our mathematical model”, in line 410 “Similar to the model presented in ref. 6”, in line 426 “inspired by ref. 6”. In all these cases the model is actually the same! If this is not spelled out with much greater precision, the field will end up being completely confused as to which model is being simulated and how various different models are different (or not) to each other. In addition, the way the model is treated is not well integrated into the rest of the

manuscript, with the modelling section seemingly bolted on to the end of the manuscript.

> We agree that a clear description of the similarities and differences of the models is essential. We revised our manuscript to emphasize that the model we use is equivalent to that of Morgan et al., but we adjusted parameter values. In particular, we describe our parameter choice and initial conditions in detail in the Methods section.

Concerning the integration of the model with the main text, we constructed the manuscript this way on purpose, to separate clearly the data from the modeling. The data we show and the many conclusions we draw are relevant independently of the coarsening model. The model is very appealing, and our data support it, but it is quite possible that it will evolve in the future, and we thus preferred to keep the data and modeling section separate.

6. The authors use the terms “droplets” and “condensates” to describe HEI10 behaviour throughout the paper. “Foci” is more appropriate as the liquid-like properties of HEI10 clusters have yet to be fully experimentally demonstrated.

> We agree and use foci in the revised manuscript

Minor points:

7. Figure 1. Could the authors also include a representative image which shows HEI10 staining (perhaps as a supplementary figure)?

> This is a good suggestion. We provide a supplemental figure with ZYP1, HEI10, and MLH1 channels separately for the four genotypes and both sexes (Figure S1)

8. Figure 1. Figure 1 E, x-axis should be “Mb”, not “M”.

> Done.

9. Figure 2 legend. Could they explain how the “permutation analysis” was performed in the methods?

> Done.

10. Lines 169-170. It should be mentioned that CO interference has already been measured in HEI10oe C2 lines both genetically and cytologically (<https://doi.org/10.1073/pnas.1713071115>, <https://doi.org/10.1038/s41467-021-24827-w>).

> Done.

11. Lines 289-299. It needs to be mentioned that the effects of HEI10 overexpression, underexpression and heterochiasmy on coarsening have already been modelled in Arabidopsis (<https://doi.org/10.1038/s41467-021-24827-w>).

> Done.

12. Line 293: “shorter SC length in females implies stronger crossover interference”. It should be made clear that this might only apply when interference is measured genetically (see point 2).

> Done

13. Figure 6A. For the measured number of MLH1 foci per cell, why is some experimental data (from Fig 1B) missing or different? For example, male/female Col HEI10oe homo is missing, but female Col HEI10oe het is present on this plot but not in Fig 1B).

> This was a mistake in the labeling of the data points. This is corrected in the revised manuscript. Thank you for picking up on it.

14. The authors often refer to the “central element” of the SC (e.g., lines 29, 54) and its role in interference. This is a bit confusing as the central element is a specific structural feature of the SC that runs along the centre of the SC (where the N-termini of the transverse filaments overlap) and the central element of the SC in Arabidopsis has not yet been characterised. The authors should therefore just refer to the “SC” more generally, rather than specifically to the “central element of the SC”.

> Indeed, the central element of the SC is yet to be characterized in Arabidopsis. However, HEI10 does localize in the middle of the SC (Capilla et al 2021, Figure 6). To make this point and still avoid confusion with the central element, we use the wording “SC” or “central zone of the SC” in the revised manuscript.

15. Line 126. As well as mentioning that HEI10oe lines maintain heterochiasmy, but zyp1 mutants do not, they could also mention that HEI10oe lines maintain CO interference (albeit weaker) but zyp1 mutants do not.

> Done.

16. Lines 154-155. They should make it clear that the “physical size” of the chromosomes refers to the genomic size, measured in Mb.

> Done.

17. Line 228. “Female and male SC lengths differ” should be changed to “Female and male axis lengths differ”, as obviously there is no SC in the zyp1 mutant.

> Done.

18. Line 396. Why was chromosome 4 excluded from interference analysis?

> Chromosome 4 bears two rearrangements, one associated with HEI10 and a second that appears to exist in our Ler wild type (see Capilla et al. 2021). As we were afraid that it could introduce artifacts, we decided not to include this chromosome in the analysis. This is explained in the revised manuscript.

19. Figure S5. The colour of the points for Male and Female COs are the same in all four plots. Is it supposed to be this way (in which case, why have they provided the key)? Is there any way to produce a plot with the male and female COs in different colours?

> This was an error. Corrected.

Reviewer #2 (Remarks to the Author):

The paper of Durand et al. builds on earlier works showing the key role of HEI10 in limiting crossover recombination (Ziolkowski et al. 2017) and SC in the control of crossover interference, crossover assurance and heterochiasmy (Capilla-Pérez et al. 2021; France et al. 2021) in Arabidopsis. The authors show that HEI10 overexpression leads to a significant reduction in interference, as has been shown to some extent before, but for the first time they show that HEI10 overexpressing lines maintain heterochiasmy. In contrast, crossover interference and heterochiasmy completely disappear in lines with zyp1 mutation, a key SC element. The authors used a number of demanding cytological and genetic techniques that allowed them to obtain a comprehensive picture of the recombination events taking place in the analyzed lines. By combining overexpression of the HEI10 protein with a mutation in the ZYP1 gene the authors showed that it is possible to significantly break the limit for the number of class I crossovers (possibly also to some extent class II) with 20 crossovers per gamete (and above 40 COs in some cases), with a complete loss of interference, assurance and heterochiasmy.

The most general conclusion is reflected in the title, saying that both HEI10 and ZYP1 contribute to the control of meiotic crossover frequency and patterning in Arabidopsis. Moreover, the presented results constitute an important test for the recombination nodule/HEI10 coarsening model (Zhang et al. 2021; Morgan et al. 2021), which is now widely discussed in the field, providing a further support for this model. Thus, this work is of great importance to our understanding of crossover interference. Overall, I believe this is a work that significantly advances our understanding of meiosis and crossover recombination and would be of great interest for a broad audience of Nature Commun. readers. Nevertheless, one of the main results of this work needs to be experimentally validated to be sure that the conclusions drawn are correct (see below, point 1).

> We are pleased that the reviewer appreciates the work

Major points:

1.

The authors observed that crossover number in female zyp1 HEI10-oe inferred from BC1 sequencing is significantly higher than the number of MLH1 foci and hypothesized that this is due to an increase in Class II crossovers. While I consider such an explanation as the most probable, it raises another serious problem that is key to some of important conclusions of this publication: If female zyp1 HEI10-oe experiences increased class II crossovers activity, this effect may be due either to HEI10 overexpression, or the zyp1 mutation. The first of these possibilities does not seem probable, because the authors do not see a similar effect in crosses with C2 line (not carrying the zyp1 mutation). Therefore, one can conclude that the loss of heterochiasmy observed in zyp1 is apparent and results in fact from hyperactivation of class II crossovers in the female zyp1 line. The consequence of this could be a specific “opposite” heterochiasmy in the zyp1 HEI10-oe line, with a higher crossover number in female than in male meiosis.

> The reviewer hypothesizes that the loss of heterochiasmy (observed by sequencing the progeny) in zyp1 is due to a hyperactivation of the class II pathway. However, this hypothesis is not supported by the data: Very importantly, the number of MLH1 foci (that mark class I COs) is identical in males and females zyp1, both in the Col and in the Col/Ler hybrid (Figure 1B). These MLH1 counts are also coherent with the genetic CO numbers. This clearly shows that there is a genuine loss of heterochiasmy (i.e., of class I COs) in zyp1. We tried to make this point clearer in the revised manuscript.

There are many speculations which could be drawn from this explanation: This would suggest that a difference in class II crossover activity between male and female meiosis contributes to heterochiasmy, which would be equally interesting. Moreover, this would indicate that HEI10 stimulates not only ZMM pathway but also class II crossovers (which I think fits well to its predicted role in the coarsening model, i.e., protection of intermediates from DNA helicases). Perhaps it would even shift the responsibility for heterochiasmy/assurance?/interference? from SC to the chromosome axis.

> The intriguing observation, which is specific to *zyp1 HEI10oe*, is the excess of genetic CO in females compared to the MLH1 foci. It indeed seems that the class II pathway is overactivated in this context (and only in this context, not in male *zyp1 HEI10oe* and not in the single mutants). The proposal that HEI10 may stimulate class II COs in the absence of *zyp1* is very interesting and is included in the revised manuscript. The overexpression of HEI10 in the absence of ZYP1 may lead to the protection of recombination intermediates that can be eventually repaired by nucleases as crossovers. At this stage, we can only speculate on why this effect is visible only in female and not in male meiosis. One possibility is that the number of DSBs might be higher in females. The observation that more COs are produced in female meiosis compared to male in the *recq4* or *recq4 figl* mutants (Fernandes et al, PNAS 2018) may support this possibility.

*Anyway, the authors need to carry out another BC1 sequencing experiment, this time for the *mus81* mutant (in a combination of *mus81 zyp1* and/or *mus81 zyp1 HEI10-oe*; both female and male meiosis). I understand that it can be challenging as an *mus81* mutant needs to be created in *Ler* background, but I think this is required to support the conclusion about the loss of heterochiasmy in *zyp1* and *zyp1 HEI10-oe* and the role of SC in crossover sex dimorphism. I also believe that this would provide us a much deeper understanding of crossover control.*

> This is an interesting suggestion, but we think that the proposed experiment goes beyond the scope of the current manuscript. Firstly, and most importantly, this point is not central to the main conclusions of the manuscript (see above). The conclusions about the crucial roles of HEI10 and ZYP1 in regulating class I COs are not affected by the intriguing, but not key, observation of a potential class II CO increase in female *zyp1 HEI10oe*. Secondly, such an experiment would take at least 5 generations, which corresponds to ~1.5 years with Arabidopsis. Thirdly, it is unclear if the experiment would give a clear conclusion as MUS81 is probably only one of the nucleases involved in the class II pathway: the double mutant *msh4 mus81* loses only 1/3 of the crossovers observed in *msh4* (10.1111/j.1365-313X.2008.03403.x), suggesting that, like in yeast, multiple nucleases contribute to the class II pathway (10.1016/j.cell.2012.03.023). We thus suggest that such an experiment could be integrated into a future work.

2.

*Wild type Col inbreds are significantly hotter than Col/Ler hybrids, which was explained in the paper by a more effective pro-crossover HEI10 allele in Col accession. In theory, the same could apply to a difference between *zyp1* inbreds and hybrids. However, even a more dramatic difference is observed between *zyp1 HEI10-oe* inbreds and hybrids. It seems unlikely that this would result from abovementioned HEI10 allele activity, since the HEI10 overexpression is close to saturation (line 107). Can the authors propose why Col/Ler hybrids show lower CO in this genetic context than Col inbreds? I think that the results presented on Fig. 4 exclude the possibility that this is linked to differences in chromosome axis length between inbreds and hybrids (as there are no differences). Is it possible that in the context of the absence of SC and the limited contact between homologous chromosomes (pairing but no synapsis), polymorphism between homologues restricts recombination to a greater*

extent than it does in WT, when SC is fully functional? To verify this hypothesis, the authors should conduct a more in-depth analysis of CO distribution against the polymorphism pattern along the chromosomes. The method of carrying out such an analysis is left to the authors' discretion.

Judging from the CO distribution maps (especially in female meiosis) in Figs. 2A and S3, certain chromosomal regions show much greater crossover suppression in *zyp1* HEI10-oe than WT, e.g. regions around 22 and 25 Mb on chromosome 1. Could the authors investigate what is so special in these regions?

> This is a very good point that we did not discuss in the original manuscript. We propose the hypothesis (included in the revised manuscript) that less CO-eligible recombination intermediates could be present in the hybrid compared to the pure Col, notably because some DSBs might be repaired on the sister because the homologous template could be too divergent in the homologous chromosome. It is possible that this difference becomes limiting only in contexts where the CO number is very high.

Concerning the variation of recombination rate along chromosomes, it should be noted that as the number of sequenced samples is limited (e.g., n=138 for female *zyp1* HEI10oe), local variation should be interpreted with caution. Nevertheless, we explored the possible cause of such variations. Notably, we compared CO distribution with gene density, SNP density and loss of synteny frequency (inversion/translocation) (Figure S5). The crossover suppression observed on some chromosomal regions appears to be frequently associated with a local loss of synteny.

Of course, other explanations for inbred/hybrid difference are also possible including different activity of Class II crossover pathways in Col and Ler accessions, and this again would be tested with the experiment proposed by me in point 1.

> We observed inbred/hybrid difference using MLH1 counting, which is specific to class I COs, and thus cannot be explained by differences in the activity of the Class II CO pathway.

3.

I like the part about the HEI10 coarsening model which I think is very clear and easy to understand. However, in Fig. 6 I miss the application of the model to the combination of *zyp1* HEI10-oe. Can the authors complete the figure with this genetic background?

> We now include *zyp1* HEI10-oe in Figure 6.

Minor comment:

- Fig. 1B and 1D I think it would be useful to show on the plot the statistical significance between male and female meiosis for each genotype.

> We agree and included this analysis in the revised Figure 1.

Reviewer #3 (Remarks to the Author):

*The data presented by Durand et al builds on recent publications in Arabidopsis that crossover interference is abolished in the *zyp1* mutant and that over-expressing HEI10 increases crossovers. It is of very high quality and of great interest to the meiosis field. The main finding is that combining the *zyp1* mutants and HEI10 overexpressing lines increase class I crossovers in Arabidopsis, determined by cytological markers such as MLH1/HEI10 and sequencing products from col/ler crosses. This is*

different to the publications on individual genes as it indicates that there is a finite pool of HEI10 protein that limits the numbers of crossovers in wild type and that this protein needs to be distributed to all chromosome pairs to maintain the obligate crossover. The authors consider that the HEI10 coarsening model explains the observed data, but it would be useful for the reader if the authors explain why other models are discounted. For example, why is that ZYP1 does not prevent closely spaced HEI10 foci and why is PCH2 not important in CO interference as suggested by Yang et al 'Bipartite recruitment of PCH2....'?

> Several models are indeed proposed to account for CO interference, but the coarsening model has the advantage of directly taking into account the key roles of HEI10 and ZYP1. The reason for favoring the coarsening model over the other available models is now set out in the revised manuscript.

Points to be corrected

Lines 14 and 15 are badly written and should be improved, especially as it is these sentences start this section.

> The abstract has been rewritten and shortened in response to the reviewer's comment and to follow the NC instructions.

Line 20. I know what you mean, but where is the 'H' for HEI10 in ZMM in proteins? This could be confusing for the non-expert. HEI10 as a class I CO protein is a better description.

> Done.

Line 21. 'shorter axis lengths' is a better way to write that statement

> Done.

Line 38. This is confusing because the previous sentence has mentioned the ZMM proteins and class I COs, but there is not a vast excess of DSBs in yeast compared to COs. It's considered to be about double. Therefore, this sentence requires more precision and a reference.

> Corrected.

Line 68. encodes 'a' not 'an'.

> Done.

Line 71. Transverse filament, not transverse element

> Done.

Line 89. Sometimes you use 'CO', but sometimes 'crossover', so consistency would be good throughout this manuscript.

> Done.

Line 93. It's not common to refer to genotypes in the plural, so wild type not wild types is better.

> Done.

Line 115. Transverse filament!

> Done.

Line 139. I don't follow this. If there are 14.7 CO per gamete and there are 4 gametes per meiosis then the number of MLH1 foci should be 14.7 x 4 at prophase, so why is it 30?

>The reviewer is right but bear in mind that a CO affects two chromatids. To help the reader we have added Figure S12.

Line 144. I also don't follow the logic of how you can infer that class II COs are increased by this analysis. Please explain.

> MLH1 foci, which mark specifically class I COs, predict less COs than observed genetically. This suggests that class II COs are also numerous in that context. We tried to make this reasoning clearer in the revised manuscript.

Line 146. For consistency please use 'transverse filament' for ZYP1 as it is not a central element protein per se, although the central element does form in this mutant.

> Edited.

Line 160. You've performed an 'assay', not written an 'essay'.

> Corrected.

Line 221. Ensure, not insure

> Corrected.

Line 231. Axes not axis

> Corrected.

Line 318. Evidence not evidences

> Corrected.

References need correcting

>checked

Reviewers' Comments:

Reviewer #1:

Remarks to the Author:

The authors have comprehensively revised their manuscript and I'm now happy for it to be published in Nature Comms. There is just one minor remaining issue in that the manuscript still mentions "gametes" (e.g. line 209) which needs altering to "transmitted chromatid".

Reviewer #2:

Remarks to the Author:

I had a chance to read the revised version of the manuscript and the author's responses to the review.

In my first review, my main comment concerned the observed difference in the frequency of male and female crossover in the background *zyp1* HEI10-oe observed by MLH1 foci and progeny sequencing. I proposed that an additional experiment using the *mus81* mutant could help to understand the reasons of this discrepancy. In their response, the authors gave a few arguments that convinced me that this experiment was not necessary for the acceptance of the paper. In particular, the inconsistency is only seen in the HEI10oe *zyp1* line, but not in the *zyp1* mutant. Therefore, I resign from the experiment suggestion made in the first review as not critical and difficult to do due to time constraints.

At the same time, I believe that the term "heterochiasmy" refers to the differences between the sexes in the number of all crossover events, and not just class I events. Therefore, I believe the authors should emphasize more in this paper that their research on the *zyp1* HEI10oe line revealed the existence of a second component heterochiasmy in Arabidopsis, which depends on class II events. While this result is presented in the paper, the authors do not refer to it as a component of heterochiasmy.

My other comments have been incorporated in the revised manuscript and I have no further objections.

Minor comments:

1. I noticed that while the authors added statistical test results for Fig. 1B, such results are not available for Fig. 1D. Does it mean that the sex differences in 1D are not statistically significant?
2. Supplementary Figures are not numbered in accordance with their order of appearance in the paper, which makes them difficult to track. Please correct the numbering accordingly.

Reviewer #3:

Remarks to the Author:

The authors have done an excellent job in responding to the reviewers comments. The manuscript is now more accessible to the reader and much better written. There is one small grammatical error on line 164 that should be corrected from 'An higher density' to 'A higher density', but as far as I am concerned the authors have dealt with my concerns.

Reviewer #1 (Remarks to the Author):

The authors have comprehensively revised their manuscript and I'm now happy for it to be published in Nature Comms. There is just one minor remaining issue in that the manuscript still mentions "gametes" (e.g. line 209) which needs altering to "transmitted chromatid".

> We have edited the manuscript to replace "gametes" by "transmitted chromatid" wherever appropriate.

Reviewer #2 (Remarks to the Author):

I had a chance to read the revised version of the manuscript and the author's responses to the review.

In my first review, my main comment concerned the observed difference in the frequency of male and female crossover in the background *zyp1* HEI10-oe observed by MLH1 foci and progeny sequencing. I proposed that an additional experiment using the *mus81* mutant could help to understand the reasons of this discrepancy. In their response, the authors gave a few arguments that convinced me that this experiment was not necessary for the acceptance of the paper. In particular, the inconsistency is only seen in the HEI10oe *zyp1* line, but not in the *zyp1* mutant. Therefore, I resign from the experiment suggestion made in the first review as not critical and difficult to do due to time constraints.

At the same time, I believe that the term "heterochiasmy" refers to the differences between the sexes in the number of all crossover events, and not just class I events. Therefore, I believe the authors should emphasize more in this paper that their research on the *zyp1* HEI10oe line revealed the existence of a second component heterochiasmy in Arabidopsis, which depends on class II events. While this result is presented in the paper, the authors do not refer to it as a component of heterochiasmy.

> We have modified the corresponding paragraph of the manuscript to refer to the new component of heterochiasmy:

„While the number of class I CO (MLH1 foci) appear to be identical in both sexes in *zyp1* HEI10oe, a novel component of heterochiasmy is revealed in this context, with now more CO in females than males, presumably due to a large increase of class II CO in female meiosis.“

My other comments have been incorporated in the revised manuscript and I have no further objections.

Minor comments:

1. I noticed that while the authors added statistical test results for Fig. 1B, such results are not available for Fig. 1D. Does it mean that the sex differences in 1D are not statistically significant?

> We have added statistical test results for figure 1D. The sex differences are statistically significant.

2. Supplementary Figures are not numbered in accordance with their order of appearance in the paper, which makes them difficult to track. Please correct the numbering accordingly.

> We have renumbered the Supplementary figures to solve this problem

Reviewer #3 (Remarks to the Author):

The authors have done an excellent job in responding to the reviewers comments. The manuscript is now more accessible to the reader and much better written. There is one small grammatical error on line 164 that should be corrected from 'An higher density' to 'A higher density', but as far as I am concerned the authors have dealt with my concerns.

> corrected